# ASAP: Exploiting the Satisficing Generalization Edge in Neural Combinatorial Optimization

**Han Fang** [1]  **Paul Weng** [2]  **Yutong Ban** [1]

## Abstract

Deep Reinforcement Learning (DRL) has emerged as a promising approach for solving Combinatorial Optimization (CO) problems, such as the 3D Bin Packing Problem (3D-BPP), Traveling Salesman Problem (TSP), or Vehicle Routing Problem (VRP), but these neural solvers often exhibit brittleness when facing distribution shifts. To address this issue, we uncover the Satisficing Generalization Edge, which we validate both theoretically and experimentally: identifying a set of promising actions is inherently more generalizable than selecting the single optimal action. To exploit this property, we propose Adaptive Selection After Proposal (ASAP), a generic framework that decomposes the decision-making process into two distinct phases: a proposal policy that acts as a robust filter, and a selection policy as an adaptable decision maker. This architecture enables a highly effective online adaptation strategy where the selection policy can be rapidly fine-tuned on a new distribution. Concretely, we introduce a two-phase training framework enhanced by Model-Agnostic Meta-Learning (MAML) to prime the model for fast adaptation. Extensive experiments on 3D-BPP, TSP, and CVRP demonstrate that ASAP improves the generalization capability of state-of-the-art baselines and achieves superior online adaptation on out-of-distribution instances.

## 1. Introduction

The domain of Combinatorial Optimization (CO) encompasses a vast array of problems that are fundamental to efficient industrial operations, logistics, and supply chain management. Among these, the 3D Bin Packing Problem

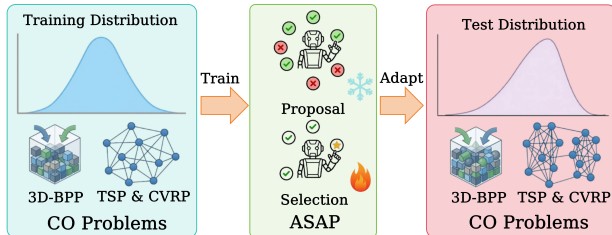

*Figure 1.* Training on fixed distributions, ASAP allows us to quickly adapt to other distributions for Combinatorial Optimization problems.

(3D-BPP), the Traveling Salesman Problem (TSP), and the Capacitated Vehicle Routing Problem (CVRP) stand as quintessential challenges (Toth & Vigo, 2014). In the 3D-BPP, the objective is to pack items of varying shapes into a container to maximize space utilization (Martello et al., 2000). Similarly, TSP and CVRP require finding optimal routes for agents to visit a set of nodes, minimizing total distance under specific constraints.

Traditionally, these NP-hard problems have been addressed using exact solvers or handcrafted heuristics. While exact solvers (e.g., Concorde (Applegate et al., 2006), Gurobi (Gurobi Optimization, LLC, 2023)) guarantee optimality, their computational cost scales exponentially, making them unsuitable for large-scale or real-time applications. Conversely, traditional heuristics (e.g., LKH-3 (Helsgaun, 2017)) offer speed but lack flexibility regarding cross-problem adaptation. To address this, Deep Reinforcement Learning (DRL) has emerged as a promising alternative, enabling the learning of heuristic solvers (policies) directly from data (Vinyals et al., 2015; Bello et al., 2017). Unlike heuristics, a DRL approach facilitates seamless adaptation by allowing the reuse of the exact same neural network architecture and training scheme across different problem environments, requiring only a retraining phase. Consequently, these neural solvers have demonstrated impressive performance on standard benchmarks, particularly for 3D-BPP (Hu et al., 2017), TSP, and CVRP (Kool et al., 2019; Nazari et al., 2018).

However, a significant limitation hinders the practical deployment of DRL-based solvers: their fragility in the face of distribution shift (Joshi et al., 2021). Neural policies

[1]Global College, Shanghai Jiao Tong University, Shanghai, China [2]Duke Kunshan University, Jiangsu, China. Correspondence to: Yutong Ban <yban@sjtu.edu.cn>.

*Proceedings of the 43rd International Conference on Machine Learning*, Seoul, South Korea. PMLR 306, 2026. Copyright 2026 by the author(s).

trained on specific data distributions (e.g., uniform node locations in TSP, or specific item sizes in 3D-BPP) often perform poorly when tested on new instances with different characteristics. This issue is pervasive in real-world scenarios where logistics patterns evolve dynamically due to seasonality, changing customer bases, or operational shifts. Consequently, two critical goals must be addressed: generalization to unseen distributions and adaptation, the ability to rapidly fine-tune a policy to a new test distribution online.

To address distribution shifts in Combinatorial Optimization (CO), we make a key observation that we call Satisficing Generalization Edge, which posits that identifying a set of promising actions generalizes better than selecting a single optimal one. For instance, in the context of TSP, identifying a subset of viable neighbors proves robust across distributions, whereas pinpointing the exact optimal action remains highly sensitive to distributional patterns. We empirically validate this observation, which suggests first selecting a proposal action set and then selecting the final action among them. We prove that this two-stage approach can be superior to the usual single-stage approach and investigate its properties, which suggests a simple training process for rapid adaptation to new test distributions.

Building upon the Satisficing Generalization Edge, we propose a generic framework called Adaptive Selection After Proposal (ASAP). Our approach fundamentally decomposes the solver's decision-making process into two distinct policies: a *proposal policy* and a *selection policy*. By decoupling these roles, the proposal policy functions as a robust filter for viable candidates, while the selection policy acts as a lightweight, adaptable decision maker. This decomposition enhances robustness and facilitates efficient online adaptation, as the selection policy can rapidly fine-tune to new distributions within the reduced action space. To effectively train this architecture, we introduce a framework combining pre-training and post-training, enhanced by Model-Agnostic Meta-Learning (MAML) (Finn et al., 2017). Overall, our contributions can be summarized as follows:

- We propose the Satisficing Generalization Edge to address the cross-distribution generalization limitations of monolithic DRL policies in CO, validating it both theoretically and empirically.

- We propose ASAP, a generic DRL framework that decouples decision-making into proposal and selection phases, supported by a meta-learning-based training scheme designed to maximize adaptability.

- We demonstrate that ASAP is a generic solution for CO. Extensive experiments on 3D-BPP, TSP, and CVRP show that ASAP outperforms state-of-the-art baselines in generalization and achieves superior online adaptation on out-of-distribution instances.

## 2. Related Work

The field of combinatorial optimization (CO) has undergone a paradigm shift with the advent of Neural Combinatorial Optimization (NCO), transitioning from hand-crafted heuristics to data-driven solvers (Bengio et al., 2021; Mazyavkina et al., 2021). We categorize recent advances into three clusters: Routing Problems, 3D Bin Packing, and Two-Stage Decision Models, highlighting the evolution from autoregressive architectures to foundation and diffusion models.

### 2.1. 3D Bin Packing Problem (3D-BPP)

Research in 3D-BPP has progressed from heuristic selection to continuous space reasoning. Early learning-based methods (Verma et al., 2020; Zhao et al., 2021) selected among heuristics like Corner or Extreme Points (Martello et al., 2000; Crainic et al., 2008; Parreño et al., 2008). This evolved into continuous packing with the Packing Configuration Tree (PCT) (Zhao et al., 2022), which allows DRL agents to evaluate heuristic-generated candidates.

**Robustness and Stability.** To handle unknown item sequences, AR2L (Pan et al., 2023b) employs adversarial attacks during training to improve worst-case robustness, while GOPT (Xiong et al., 2024) uses Transformers to encode spatial correlations for dimensional generalization. Physical realizability is ensured by frameworks like One4Many-StablePacker (Gao et al., 2025b), which integrates stability validation, and physics-based pipelines (Zhao et al., 2023; Song et al., 2023) that simulate interactions for irregular shapes and retrieval tasks. Different from methods focusing solely on robust encoders or adversarial training, we propose a proposal-selection mechanism that enables rapid online adaptation.

### 2.2. Routing Problems (TSP/CVRP)

Routing problems like TSP and CVRP remain the primary testbed for NCO, including search-based methods (Ouyang et al., 2021a;b; Fu et al., 2021), autoregressive methods (Dai et al., 2017; Jin et al., 2023) and Divide-conquer methods (Hou et al., 2023; Pan et al., 2023a; Ye et al., 2023). The dominant *constructive* approach originated with Pointer Networks (Vinyals et al., 2015) and was refined by the Attention Model (AM) (Kool et al., 2019), which established a Transformer-based baseline. To address the instability of REINFORCE training, POMO (Kwon et al., 2020) introduced instance symmetries and multiple greedy rollouts.

**Generalization and Adaptation.** A critical bottleneck is the "generalization gap" on large-scale or out-of-distribution instances. While architectures like LEHD (Luo et al., 2023) address this challenge by shifting complexity to the decoder, other generalization strategies employ different mechanisms. These include ensembles leveraging transferrable

local policies (Gao et al., 2023), invariant nested encoders (INVIT) (Fang et al., 2024) designed to mitigate distant node interference, and Multi-View Graph Contrastive Learning (MVGCL) (Jiang et al., 2023) to distill transferable patterns. Several studies investigate adaptation strategies, including employing meta-learning initializations such as Omni-TSP (Zhou et al., 2023). Recently, Large Language Models (LLMs) have been employed as meta-optimizers for test-time projection (TTPL) (Chen et al., 2025) or to guide heuristic evolution (Chi et al., 2026).

**Foundation and Generative Models.** To address the generalization challenges of CO, another stream of research leverages Foundation and Generative Models. RouteFinder (Berto et al., 2025) solves 48 VRP variants via modular attributes, and MVMoE (Zhou et al., 2024) utilizes Mixture-of-Experts for multi-task learning. Beyond construction, diffusion models like DIFUSCO (Sun & Yang, 2023) and IDEQ (Basson & Preux, 2024) generate solutions via iterative denoising of edge heatmaps, while Adversarial Generative Flow Networks (AGFN) (Zhang et al., 2025) combine GFlowNets with adversarial training to escape local optima. In contrast to them, our framework applies the Satisficing Generalization Edge by decoupling decision-making, enhancing cross-distribution generalization.

### 2.3. Two-Stage Decision Models

State-of-the-art two-stage models typically hybridize Machine Learning with traditional optimization to manage combinatorial explosion. In contrast, ASAP presents a purely Machine Learning-based two-stage framework.

**Predict-and-Search.** These frameworks predict solution properties to guide exact solvers. GNN-guided approaches (Han et al., 2023) define trust regions for MILPs, while ConPaS (Huang et al., 2024) and ID-PaS (Cai et al., 2025) use contrastive and identity-aware learning to refine variable predictions and partial assignments.

**Pruning and Selection.** "Learning to Prune" (L2P) identifies unpromising search spaces. Theoretical foundations for pruning repeated computations (Alabi et al., 2019) have been applied to the Steiner Tree problem (Zhang & Ajwani, 2024) and constructive routing (Zhou et al., 2025) to reduce the decoder's burden. Alternatively, Neural Solver Selection (Gao et al., 2025a) treats algorithm choice as the first optimization stage, leveraging a portfolio of pre-trained solvers. Generative-discriminative architectures, such as the proposal-selection split seen in AGFN (Zhang et al., 2025), allow for separating candidate diversity from rigorous selection. Unlike these approaches which largely aid exact solvers or select pre-trained agents, our method integrates proposal and selection into a unified DRL framework to maximize few-shot adaptability.

## 3. Background

In this section, we provide the formal definitions of the combinatorial optimization problems and subsequently reformulate them as sequential decision-making tasks.

### 3.1. Problem Definitions

The Traveling Salesperson Problem (TSP) and the Capacitated Vehicle Routing Problem (CVRP) are defined on a graph $G = (V, E)$, where $V = \{1, \ldots, n\}$ represents the set of nodes and edges $(i, j) \in E$ have associated costs $c_{ij}$. In the TSP (Applegate et al., 2006), the goal is to find a permutation $\pi$ of nodes that minimizes the total tour length $\sum_{i=1}^{n-1} c_{\pi(i),\pi(i+1)} + c_{\pi(n),\pi(1)}$, visiting every node exactly once. The CVRP (Dantzig & Ramser, 1959) generalizes the TSP by introducing a specific depot node $0$, customer demands $d_i > 0$ for each node $i \in V \setminus \{0\}$, and a fleet of vehicles with uniform capacity $Q$. Unlike the TSP, where a single tour visits all nodes, the CVRP requires partitioning nodes into multiple routes $\mathcal{R}_1, \ldots, \mathcal{R}_K$ such that each route starts and ends at the depot and satisfies the capacity constraint $\sum_{i \in \mathcal{R}_k} d_i \leq Q$. The objective shifts to minimizing the aggregate travel cost across all vehicle routes. Following previous work in NCO, we focus on Euclidean TSP and CVRP (i.e., nodes are positioned in the unit square $[0, 1]^2$ and edge costs correspond to Euclidean distances).

The Online 3D Bin Packing Problem (3D-BPP) addresses the challenge of packing a sequence of rectangular items $\mathcal{I}$ into a fixed container of dimensions $(L, W, H)$ (Martello et al., 2000; Hu et al., 2017). Each item $i$ is defined by its dimensions $(l_i, w_i, h_i)$. In the online setting, items arrive sequentially, and the decision-maker must pack the current item $i$ at a coordinate $(x_i, y_i, z_i)$ with an orientation $o_i$ before observing the subsequent items. Constraints dictate that all packed items must be strictly contained within the bin boundaries and must not overlap. The optimization objective is to maximize the Space Utilization Ratio (SUR), defined as $\frac{\sum_{i \in \mathcal{P}} l_i w_i h_i}{L \times W \times H}$, where $\mathcal{P} \subseteq \mathcal{I}$ represents the subset of successfully packed items.

### 3.2. Sequential Decision Making Formulations

We formulate the TSP and CVRP as Markov Decision Processes (MDPs), characterized by the tuple $(\mathcal{S}, \mathcal{A}, P, R)$, because the full graph structure is observable (Bello et al., 2017; Kool et al., 2019). The state $s_t$ tracks the partial solution: for TSP, this includes the current location and visited nodes, while the CVRP also tracks the accumulated vehicle load (Nazari et al., 2018). The action space $\mathcal{A}$ consists of choosing the next node or returning to the depot. The transition $P$ is deterministic, updating the current location and visited set upon action execution. The reward $R(s_t, a_t)$ is defined as the negative edge weight $-c_{ij}$, aligning with the

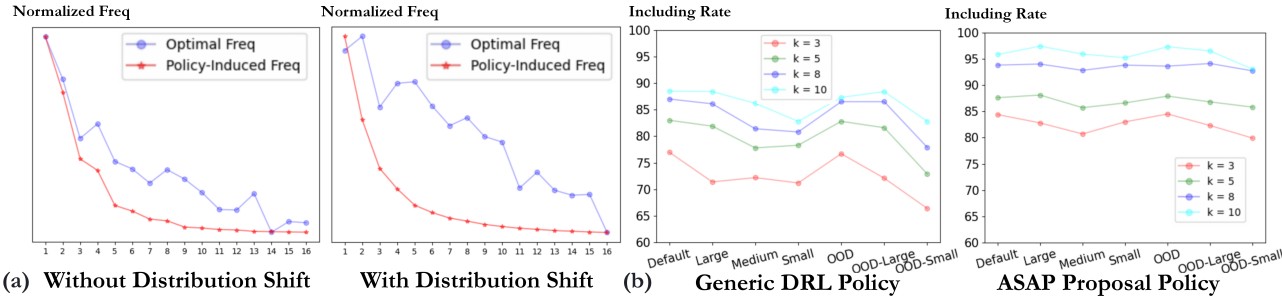

*Figure 2.* Preliminary results (see Section 5.1 for dataset description) indicate the factor leading to the generalization gap. (a) Comparison of Optimal Frequencies and Policy-Induced Frequencies vs. Rank of Choices. The left figure shows the results on the Default (training) dataset, while the right figure displays the results on the ID-Small dataset. (b) Results of top-1 action (induced by MCTS) including rate by different sizes of proposal action set on cross-distribution datasets.

minimization of total tour cost.

Conversely, we model the Online 3D-BPP as a Partially Observable Markov Decision Process (POMDP) due to uncertainty regarding future items (Zhao et al., 2022). While the true state $\mathcal{S}$ includes the full sequence of incoming items, the agent only receives an observation $o_t \in \Omega$ comprising the current bin configuration (e.g., a height map or voxel grid) and the dimensions of the immediate next item. The action $a_t$ determines the placement position and orientation. The transition function $T$ deterministically updates the bin state and reveals the next item in the sequence, while the reward $R$ is based on the item's volume $l_i \times w_i \times h_i$ to encourage packing efficiency.

## 4. Method

We begin this section by formalizing our key insights, Satisficing Generalization Edge, via two theorems alongside empirical validations. As a practical application of these theoretical advancements, we introduce Adaptive Selection After Proposal (ASAP). This generic framework is designed for combinatorial optimization tasks, including the 3D Bin Packing Problem (3D-BPP) and Routing Problems (TSP/CVRP). Subsequently, we detail the ASAP architecture and the associated two-phase training scheme.

### 4.1. Two-stage Decision Process

Under distribution shift, a trained policy $\pi$ often suffers performance degradation, failing to pinpoint the optimal action in out-of-distribution instances. However, the optimal action frequently remains within the policy's top-$k$ candidates. This motivates the *Satisficing Generalization Edge*, which posits that identifying promising actions is inherently more generalizable than selecting a best action. To empirically validate this claim, we conduct preliminary experiments on 3D-BPP (see Appendix C for details), employing Monte-Carlo Tree Search (MCTS) to approximate the optimal policy. In this setting, a trained policy $\pi$ generates a proposal set comprising the top-$k$ actions by probability.

First, we investigate the validity of the proposal set by using the policy to prune actions for the MCTS search. As shown in Figure 2 (b), even with a small proposal size $k$ (e.g., top-3), the optimal action determined by MCTS is included in the policy's proposal set with a high rate (over 65%) across cross-distribution datasets. Consequently, limiting the search space to these proposed actions does not result in a significant performance drop, confirming the robustness of the proposal set. Second, we compare the frequency distributions. The MCTS induces an *Optimal Frequency* by consistently solving instances to mitigate sampling randomness, while the policy generates a *Policy-Induced Frequency*. The comparison in Figure 2 (a) reveals that these frequencies match globally (sharing a decreasing trend) but mismatch locally on out-of-distribution datasets. This indicates that while the policy captures the global "promising" landscape, it struggles with the precise local ranking.

Since a trained policy $\pi$ can more effectively identify promising actions than an optimal one, we define a two-stage decision process and theoretically analyze its superiority in Theorem 4.2:

**Definition 4.1** (Two-Stage Decision Process). The two-stage decision process selects an action from the action set $\mathcal{A}$ with size $n$ through two sequential steps:

**Stage 1 (Proposal):** A proposal set $\hat{\mathcal{A}} \subseteq \mathcal{A}$ of size $k \ll n$ is sampled by policy $\pi^p$ and $\mathbb{P}^{pro} = \mathbb{P}(a_{t_1} \in \hat{\mathcal{A}})$ denote the probability of including the optimal action $a_{t_1}$.

**Stage 2 (Selection):** A final action $a$ is sampled from the proposal set $\hat{\mathcal{A}}$ and $\mathbb{P}^{sel} = \mathbb{P}(a_{t_1} \sim \pi^s(\cdot \mid s) \mid a_{t_1} \in \hat{\mathcal{A}})$ denote the probability of selecting the optimal action $a_{t_1}$.

To simplify the theoretical analysis, , our proof assumes a worst-case scenario where the policies for proposal and selection steps are identical to one-step decision process ($\pi = \pi^p = \pi^s$). In practice, cooperative tuning guarantees performance that exceeds this theoretical lower bound.

**Theorem 4.2** (Superiority of Two-Stage Decision Process). *Given a policy $\pi$ assigning probability $p_{t_1}$ to the optimal action, the probability of selecting the optimal action via the*

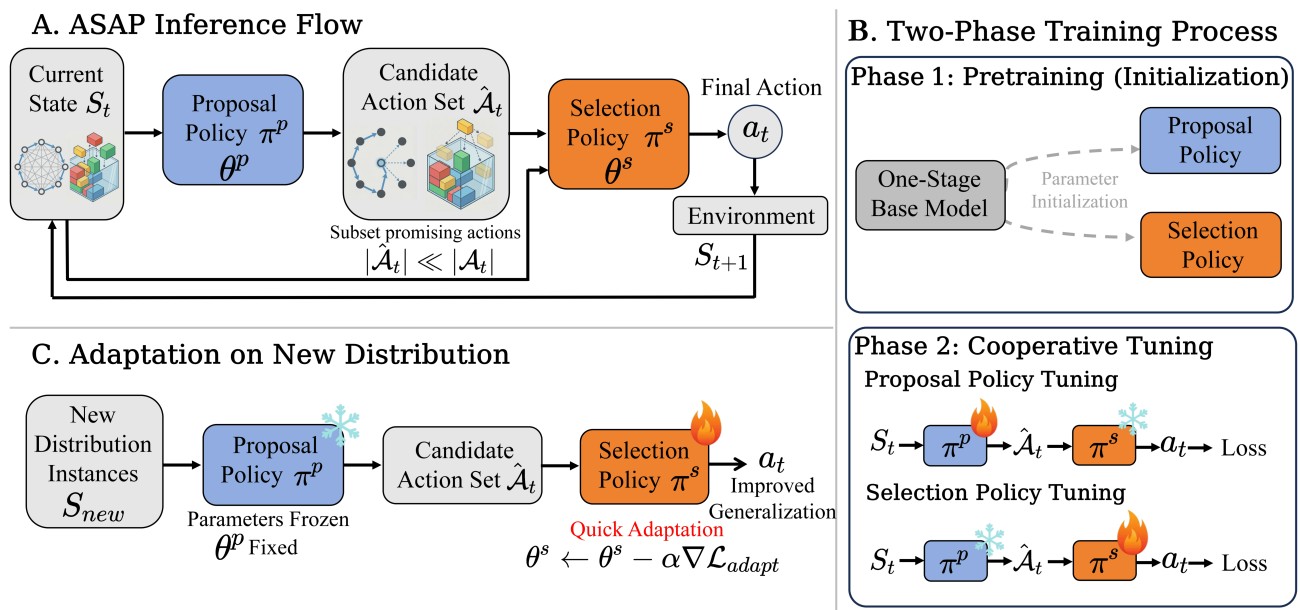

*Figure 3.* Overview of the ASAP architecture. (A) **Inference Flow**: The decision process is decoupled into a Proposal Policy ($\pi^p$) that generates a candidate subset, and a Selection Policy ($\pi^s$) that picks the final action. (B) **Two-Phase Training**: To ensure convergence, we first pretrain a base model (Phase 1) to initialize parameters, followed by cooperative tuning (Phase 2) where both policies interact. (C) **Adaptation**: On new distributions, the proposal policy is frozen to maintain general candidate quality, while the selection policy is quickly fine-tuned by interacting with the environment to fit the specific domain.

*Two-Stage process ($\mathbb{P}^{two}$) strictly exceeds the probability via the One-Stage process ($\mathbb{P}^{one}$), i.e., $\mathbb{P}^{two} > \mathbb{P}^{one}$, if:*

$$p_{t_1} > \frac{1}{n-1}.$$

Theorem 4.2 provides a theoretical guarantee that the two-stage process outperforms the one-stage process provided the policy is better than random guessing. In addition, we analyze an interesting property in this two-stage process:

**Theorem 4.3** (Robustness of Proposal Set Inclusion). *Let $\pi$ be a policy assigning probability $p_{t_1}$ to the optimal action $a_{t_1}$, and let $\pi^*$ be the optimal policy assigning probability $p_{t_1}^*$ to the same action. If the proposal set size $k$ exceeds a threshold $k_0$ given by:*

$$k > k_0 \approx \frac{\ln\left((n-1)p_{t_1}^* p_{t_1}\right)}{p_{t_1}}$$

*then we have:*

$$\mathbb{P}^{pro} > \mathbb{P}^{sel}, \quad \frac{\partial \mathbb{P}^{pro}}{\partial p_{t_1}} < \frac{\partial \mathbb{P}^{sel}}{\partial p_{t_1}}$$

A detailed proof of Theorem 4.2 and Theorem 4.3 is provided in Appendix B. Theorem 4.3 suggests a sufficient strategy for domain adaptation: decouple the proposal and selection process, and only fine-tune the selection policy. Intuitively, by constraining the search to a small set of promis-

ing candidates, the selection policy operates within a significantly reduced action space ($k \ll n$), thereby accelerating adaptation to new distributions.

### 4.2. ASAP

We propose ASAP, a framework designed to improve decision-making efficiency and adaptability by decoupling the action generation process. Figure 3 illustrates the overall architecture of our method, covering the inference flow, the training strategy, and the adaptation mechanism.

**ASAP Inference Flow.** As shown in Figure 3(A), the core mechanism of ASAP operates by splitting the decision-making process into two distinct steps: proposal and selection. Given a current state $S_t$, the Proposal Policy ($\pi^p$, parameterized by $\theta^p$) first generates a probability distribution over the entire action space. Based on this distribution, it outputs a Candidate Action Set $\hat{\mathcal{A}}_t$, which contains a subset of the most promising actions such that $|\hat{\mathcal{A}}_t| \ll |\mathcal{A}_t|$. Subsequently, the Selection Policy ($\pi^s$, parameterized by $\theta^s$) takes this reduced set $\hat{\mathcal{A}}_t$ as input. It evaluates the candidates and samples a Final Action $a_t \sim \pi^s(\cdot|S_t, \hat{\mathcal{A}}_t)$ to interact with the environment. By filtering out irrelevant actions early, the selection policy can focus its computational resources on distinguishing between high-quality candidates instead of randomly exploring other actions.

**Training Challenges.** While the decoupled architecture offers inference efficiency, training the proposal and selection policies cooperatively from scratch is non-trivial. The two policies are interdependent: a suboptimal proposal policy may exclude the ground-truth optimal action from the candidate set $\hat{\mathcal{A}}_t$. In such cases, even a perfect selection policy cannot recover the optimal action, leading to a "garbage-in, garbage-out" scenario. Conversely, an untrained selection policy provides noisy feedback to the proposal policy. This mutual interference often causes the training process to diverge or get trapped in poor local optima if both are initialized randomly.

**Two-Phase Training.** To address the convergence difficulties, we introduce a Two-Phase training process, as illustrated in Figure 3(B).

- **Phase 1: Pretraining (Initialization).** Instead of training the decoupled policies immediately, we start with a *One-Stage Parameter Base Model Initialization*. We train a standard policy on the environment to learn a general understanding of the state-action representation. The weights from this base model are then used to initialize both the selection policy parameters $\theta^s$ and the proposal policy parameters $\theta^p$. This provides a "warm start," ensuring that the cooperative training begins with a reasonable policy.

- **Phase 2: Cooperative Tuning.** Once initialized, the system enters the cooperative tuning stage. Here, the policies operate in their decoupled roles. The proposal policy $\pi^p$ generates the candidate set $\hat{\mathcal{A}}_t$, and the selection policy $\pi^s$ samples action $a_t$. Both policies are updated based on the interaction experience. The loss is backpropagated to refine both $\theta^p$ and $\theta^s$, allowing them to adapt to each other—the proposal policy learns to include actions that the selection policy prefers, while the selection policy learns to pick the best option from the provided candidates.

**Quick Adaptation on New Distributions.** A key advantage of our framework is its ability to adapt efficiently to new environments, as depicted in Figure 3(C). When the model encounters instances from a New Distribution, fully retraining the entire architecture is computationally expensive and slow. We observe that the criteria for "promising candidates" (handled by the Proposal Policy) are generally universal across distributions, whereas the specific "optimal choice" (handled by the Selection Policy) is more domain-sensitive. Therefore, during the adaptation phase, we keep the proposal policy parameters frozen ($\theta^p$ Fixed). We only update the selection policy via gradient descent: $\theta^s \leftarrow \theta^s - \alpha \nabla \mathcal{L}_{adapt}$. Crucially, $\mathcal{L}_{adapt}$ is not a newly proposed loss function but directly inherits the standard loss function of the specific DRL baseline it is integrated with. Because the selection policy operates on a significantly reduced action space ($\hat{\mathcal{A}}_t$), it requires fewer samples to converge. This allows ASAP to achieve improved performance on $S_{new}$ with rapid adaptation steps.

**Additional Strategy for Improving ASAP.** The performance of ASAP can be further enhanced by incorporating auxiliary strategies, such as meta-learning, to improve the model's initialization and adaptability. We specifically employ Model-Agnostic Meta-Learning (MAML) (Finn et al., 2017) for this purpose, as it is easy to implement and highly effective for generalization tasks. To implement MAML within the context of combinatorial optimization, we treat each distinct data distribution as an independent task. During the training process, we sample various distributions $p_i(\mathcal{I})$ of the input item set $\mathcal{I}$. For each sampled distribution $p_i(\mathcal{I})$, we first compute task-specific adapted weights $\theta_i'$ by solving a batch of instances $X_i$ drawn from $p_i(\mathcal{I})$, starting from the initial policy weights $\theta$. Subsequently, we collect new trajectories by solving these instances with the adapted weights $\theta_i'$. The global policy weights $\theta$ are then updated by learning from these trajectories, thereby producing an initialization that can quickly adapt to diverse problem distributions. More details in Appendix D.

## 5. Experimental Results

To validate the effectiveness of our proposed method, we conduct comprehensive evaluations on two distinct domains: the Routing Problem (TSP and CVRP) and the Online 3D Bin Packing Problem (3D-BPP). We utilize datasets featuring various scales and distributions to assess performance under distribution shifts and scaling challenges.

### 5.1. Experimental Setup

**Datasets.** To evaluate cross-distribution and cross-size generalization, we employ distinct protocols for routing and packing problems. For TSP and CVRP, we generate instances using the Multi-Scale Various-Distribution Routing Problem (MSVDRP) protocol (Bossek et al., 2019b). This includes four node distributions (*Uniform*, *Clustered*, *Explosion*, *Implosion*) across two scales: small (TSP-100, CVRP-50) and large (TSP-1000, CVRP-500).For Online 3D-BPP, we utilize a standard container size of $20^3$ with both discrete and continuous item dimensions. We construct In-Distribution (ID) and Out-of-Distribution (OOD) datasets containing varying item scales and unseen dimension ranges to rigorously test generalization. Specific generation parameters and instance counts are detailed in Appendix E. Due to space constraints, we present only the most salient results here; comprehensive results are provided in Appendix F.

**Baselines and Metrics.** For 3D-BPP, we compare with

*Table 1.* Discrete and Continuous Performance Comparison with and without Online Adaptation on 3D-BPP Datasets. Approx Optim represents the performance of MCTS.

| | Measurement | Discrete | | | | | | Continuous | | | | | |
|---|---|---|---|---|---|---|---|---|---|---|---|---|---|
| | | w/o Adaptation | | w/ Adaptation | | Improvement | Inference | w/o Adaptation | | w/ Adaptation | | Improvement | Inference |
| | | Uti(%) | Num | Uti(%) | Num | ΔUti(%) | time(m) | Uti(%) | Num | Uti(%) | Num | ΔUti(%) | time(m) |
| **ID-Large** | Approx Optim | 75.3 | 12.6 | / | / | / | / | 65.5 | 10.8 | / | / | / | / |
| | PCT(ICLR-22) | 70.0 | 10.9 | 70.2 | 11.0 | +0.2 | 1.0 | 61.4 | 9.7 | 61.4 | 9.8 | +0.0 | 5.8 |
| | AR2L(Neurips-23) | 68.8 | 10.5 | 69.3 | 10.7 | +0.3 | 1.1 | 61.2 | 9.7 | 61.5 | 9.8 | +0.3 | 5.8 |
| | GOPT(RAL-24) | 71.2 | 11.3 | 71.0 | 11.3 | -0.2 | 1.1 | / | / | / | / | / | / |
| | PCT+ASAP w/o MAML | 72.9 | 12.0 | 73.5 | 12.1 | +0.6 | 1.1 | 62.5 | 10.1 | 63.4 | 10.2 | **+0.9** | 5.9 |
| | PCT+MAML | 71.7 | 11.5 | 72.0 | 11.6 | +0.3 | 1.1 | 62.3 | 9.9 | 62.3 | 9.9 | +0.0 | 5.8 |
| | PCT+ASAP w/ MAML | **73.5** | 12.1 | **74.2** | 12.3 | **+0.7** | 1.1 | **63.1** | 10.2 | **63.9** | 10.3 | +0.8 | 5.9 |
| | PCT+ASAP w/ MAML(Tune All) | **73.5** | 12.1 | 73.7 | 12.2 | +0.2 | 1.1 | **63.1** | 10.2 | 63.4 | 10.3 | +0.3 | 5.9 |
| **ID-Small** | Approx Optim | 87.3 | 100.2 | / | / | / | / | 70.1 | 84.0 | / | / | / | / |
| | PCT(ICLR-22) | 82.9 | 94.5 | 83.4 | 95.1 | +0.5 | 12.0 | 65.0 | 79.6 | 65.4 | 80.2 | +0.4 | 42.0 |
| | AR2L(Neurips-23) | 82.1 | 93.6 | 82.5 | 94.4 | +0.4 | 12.1 | 65.2 | 79.5 | 65.7 | 80.3 | +0.5 | 42.1 |
| | GOPT(RAL-24) | 84.5 | 97.5 | 84.7 | 97.4 | +0.2 | 12.1 | / | / | / | / | / | / |
| | PCT+ASAP w/o MAML | 85.3 | 98.0 | 86.7 | 99.5 | **+1.4** | 12.2 | 67.0 | 81.0 | 68.1 | 81.8 | **+1.1** | 42.3 |
| | PCT+MAML | 85.8 | 98.2 | 86.0 | 98.3 | +0.2 | 12.0 | 66.8 | 80.8 | 66.9 | 80.8 | +0.1 | 42.0 |
| | PCT+ASAP w/ MAML | **86.5** | 99.0 | **87.4** | 101.0 | +0.9 | 12.2 | **67.6** | 81.4 | **68.3** | 81.9 | +0.7 | 42.3 |
| | PCT+ASAP w/ MAML(Tune All) | **86.5** | 99.0 | 87.0 | 100.2 | +0.5 | 12.2 | **67.6** | 81.4 | **68.3** | 81.9 | +0.7 | 42.3 |
| **OOD-Large** | Approx Optim | 65.3 | 7.4 | / | / | / | / | 60.1 | 7.1 | / | / | / | / |
| | PCT(ICLR-22) | 60.2 | 6.6 | 60.6 | 6.7 | +0.4 | 1.0 | 53.1 | 5.9 | 53.3 | 5.9 | +0.2 | 5.0 |
| | AR2L(Neurips-23) | 58.8 | 6.5 | 59.0 | 6.5 | +0.2 | 1.0 | 51.7 | 5.8 | 52.0 | 5.8 | +0.3 | 5.1 |
| | GOPT(RAL-24) | 59.9 | 6.6 | 60.1 | 6.6 | +0.2 | 1.0 | / | / | / | / | / | / |
| | PCT+ASAP w/o MAML | **61.5** | 6.7 | 62.2 | 6.9 | +0.7 | 1.0 | 54.2 | 6.1 | 56.1 | 6.3 | **+1.9** | 5.1 |
| | PCT+MAML | 60.8 | 6.7 | 61.0 | 6.7 | +0.2 | 1.0 | 54.5 | 6.1 | 54.8 | 6.2 | +0.3 | 5.0 |
| | PCT+ASAP w/ MAML | **61.5** | 6.7 | **62.4** | 6.9 | **+0.9** | 1.0 | **55.7** | 6.2 | **57.0** | 6.4 | +1.3 | 5.1 |
| | PCT+ASAP w/ MAML(Tune All) | **61.5** | 6.7 | 62.0 | 6.8 | +0.5 | 1.0 | **55.7** | 6.2 | 56.3 | 6.3 | +0.6 | 5.1 |
| **OOD-Small** | Approx Optim | 84.8 | 149.5 | / | / | / | / | 73.1 | 125.1 | / | / | / | / |
| | PCT(ICLR-22) | 79.5 | 142.4 | 80.3 | 143.0 | +0.8 | 16.8 | 67.5 | 116.1 | 68.0 | 116.9 | +0.5 | 47.2 |
| | AR2L(Neurips-23) | 78.8 | 140.0 | 79.7 | 141.8 | +0.9 | 16.9 | 68.3 | 117.5 | 68.9 | 119.0 | +0.6 | 47.3 |
| | GOPT(RAL-24) | 80.5 | 143.5 | 81.2 | 144.6 | +0.7 | 16.8 | / | / | / | / | / | / |
| | PCT+ASAP w/o MAML | 81.8 | 145.6 | 83.9 | 147.6 | +2.1 | 17.1 | 69.5 | 119.6 | 71.8 | 122.8 | **+2.3** | 47.6 |
| | PCT+MAML | 81.2 | 145.0 | 82.1 | 146.2 | +0.9 | 16.8 | 69.6 | 119.4 | 70.1 | 120.8 | +0.5 | 47.3 |
| | PCT+ASAP w/ MAML | **82.3** | 146.5 | **84.5** | 148.8 | **+2.2** | 17.1 | **70.5** | 121.3 | **72.6** | 124.2 | +2.1 | 47.6 |
| | PCT+ASAP w/ MAML(Tune All) | **82.3** | 146.5 | 83.4 | 147.3 | +1.1 | 17.1 | **70.5** | 121.3 | 71.5 | 122.2 | +1.0 | 47.6 |

online DRL-based solvers such as PCT (Zhao et al., 2022), AR2L (Pan et al., 2023b), and GOPT (Xiong et al., 2024). For routing, we integrate and compare against SOTA neural constructive solvers including POMO (Kwon et al., 2020) and INViT (Fang et al., 2024). For 3D-BPP, we report *Space Utilization* ($Uti$). Performance on routing problems is measured by the average *optimality gap* relative to exact (Gurobi (Gurobi Optimization, LLC, 2023)) or heuristic (LKH3 (Helsgaun, 2017), HGS (Vidal, 2022)) solvers.

**Implementation.** Models, including baseline models and our proposed ASAP, are trained exclusively on small-scale instances with uniform distributions (TSP-100/CVRP-50 or BPP-Default) to evaluate zero-shot generalization and performance after few-shot adaptation. Comprehensive implementation details, including hardware and hyperparameters, are provided in Appendix E.

## 5.2. Performance Analysis

**Performance on 3D-BPP.** We evaluate the performance of our proposed method on the 3D-BPP task by comparing PCT+ASAP w/o MAML against state-of-the-art baseline

methods, including PCT, AR2L, and GOPT, as shown in Table 1. In terms of **generalization performance** (before adaptation), PCT+ASAP w/o MAML consistently outperforms the baselines across all datasets. For instance, on the ID-Large (Discrete) dataset, PCT+ASAP w/o MAML achieves a utilization of 72.9%, surpassing PCT (70.0%), AR2L (68.8%), and GOPT (71.2%) by significant margins. Similarly, in the OOD-Small (Discrete) scenario, our method achieves 81.8% utilization compared to 79.5% for PCT and 80.5% for GOPT. Regarding **adaptation performance**, PCT+ASAP w/o MAML demonstrates superior online adaptation capabilities. On the ID-Large dataset, while PCT improves by only 0.2% after adaptation, PCT+ASAP w/o MAML achieves a 0.6% gain. This trend is even more pronounced in continuous settings; for OOD-Small (Continuous), our method achieves a substantial improvement of 2.3% (ΔUti), whereas PCT and AR2L only improve by 0.5% and 0.6%, respectively. See more results on other distributions in Section F.1.1.

**Performance on TSP/CVRP.** Table 2 presents the comparison on TSP and CVRP tasks, focusing on POMO and

*Table 2.* Comparison of Optimality Gaps (Pre- and Post-Adaptation) and Runtime for POMO and INViT-1V Policies w/ and w/o ASAP across Clustered, Implosion, and Explosion Datasets.

| Policy | FT: TSP-100 / Eval: TSP-100 | | | | FT: TSP-100 / Eval: TSP-1000 | | | | FT: CVRP-50 / Eval: CVRP-50 | | | | FT: CVRP-100 / Eval: CVRP-500 | | | |
|---|---|---|---|---|---|---|---|---|---|---|---|---|---|---|---|---|
| | Before | After | Imp. | Time | Before | After | Imp. | Time | Before | After | Imp. | Time | Before | After | Imp. | Time |
| **Clustered** | | | | | | | | | | | | | | | | |
| (Near-)Optimality | 0.00% | / | / | 3.4m | 0.00% | / | / | 16.9h | 0.00% | / | / | 34.2m | 0.00% | / | / | 2.3d |
| POMO | 5.22% | 5.11% | 0.11% | 0.3m | 57.25% | 68.23% | -10.98% | 2.3m | 7.53% | 8.22% | -0.69% | 0.5m | 26.80% | 24.21% | 2.59% | 3.7m |
| POMO+ASAP w/o MAML | 4.32% | 3.85% | **0.47%** | 0.3m | 47.88% | 40.01% | **7.87%** | 2.3m | 6.68% | **5.11%** | **1.57%** | 0.5m | 20.15% | 15.19% | **4.96%** | 3.7m |
| POMO+MAML | 5.07% | 4.77% | 0.30% | 0.3m | 58.11% | 59.67% | -1.56% | 2.3m | **6.65%** | 6.13% | 0.52% | 0.5m | 25.33% | 23.87% | 1.46% | 3.7m |
| POMO+ASAP w/ MAML | **4.15%** | **3.82%** | 0.33% | 0.3m | **46.15%** | **38.73%** | 7.42% | 2.3m | 6.72% | 5.33% | 1.39% | 0.5m | **19.73%** | **15.35%** | 4.38% | 3.7m |
| POMO+ASAP w/ MAML(Tune All) | **4.15%** | 4.05% | 0.10% | 0.3m | **46.15%** | 42.13% | 4.02% | 2.3m | 6.72% | 6.22% | 0.50% | 0.5m | **19.73%** | 17.42% | 2.31% | 3.7m |
| INViT-1V | 7.14% | 6.77% | 0.37% | 0.4m | 14.30% | 14.43% | -0.13% | 4.5m | 6.34% | 6.21% | 0.13% | 1.0m | 12.25% | 11.86% | 0.39% | 10.3m |
| INViT-1V+ASAP w/o MAML | 4.80% | 4.36% | 0.44% | 0.5m | **11.01%** | 8.12% | 2.89% | 5.3m | 5.51% | 4.05% | 1.46% | 1.1m | 11.89% | 9.43% | **2.46%** | 10.7m |
| INViT-1V+MAML | 6.87% | 6.35% | **0.52%** | 0.4m | 13.01% | 12.11% | 0.90% | 4.5m | 5.75% | 4.59% | 1.16% | 1.0m | 12.43% | 11.33% | 1.10% | 10.3m |
| INViT-1V+ASAP w/ MAML | **4.45%** | **4.11%** | 0.34% | 0.5m | 11.13% | **7.94%** | **3.19%** | 5.3m | **5.43%** | **3.78%** | **1.65%** | 1.1m | **11.37%** | 9.15% | 2.22% | 10.7m |
| INViT-1V+ASAP w/ MAML(Tune All) | **4.45%** | 4.32% | 0.13% | 0.5m | 11.13% | 8.22% | 2.91% | 5.3m | **5.43%** | 4.32% | 1.11% | 1.1m | **11.37%** | **9.01%** | 2.36% | 10.7m |
| **Implosion** | | | | | | | | | | | | | | | | |
| (Near-)Optimality | 0.00% | / | / | 2.8m | 0.00% | / | / | 17.5h | 0.00% | / | / | 30.0m | 0.00% | / | / | 2.0d |
| POMO | 2.53% | 2.67% | -0.14% | 0.3m | 69.93% | 74.23% | -4.30% | 2.3m | 7.93% | 7.71% | 0.22% | 0.5m | 24.32% | 26.15% | -1.83% | 3.7m |
| POMO+ASAP w/o MAML | 2.22% | 2.17% | 0.05% | 0.3m | 60.11% | 57.96% | 2.15% | 2.3m | 6.41% | 4.99% | 1.42% | 0.5m | **19.07%** | 15.22% | 3.85% | 3.7m |
| POMO+MAML | 2.31% | 2.26% | 0.05% | 0.3m | 65.74% | 64.13% | 1.61% | 2.3m | 7.17% | 6.02% | 1.15% | 0.5m | 22.23% | 21.89% | 0.34% | 3.7m |
| POMO+ASAP w/ MAML | **2.21%** | **2.05%** | 0.16% | 0.3m | **56.73%** | **52.11%** | 4.62% | 2.3m | **6.35%** | **4.61%** | 1.74% | 0.5m | 19.35% | **14.79%** | **4.56%** | 3.7m |
| POMO+ASAP w/ MAML(Tune All) | **2.21%** | 2.11% | 0.10% | 0.3m | **56.73%** | 53.32% | 3.41% | 2.3m | **6.35%** | 4.59% | **1.76%** | 0.5m | 19.35% | 16.02% | 3.33% | 3.7m |
| INViT-1V | 6.48% | 5.69% | 0.79% | 0.4m | 13.18% | 11.98% | 1.20% | 4.5m | 6.99% | 6.77% | 0.22% | 1.0m | 13.32% | 13.33% | -0.01% | 10.3m |
| INViT-1V+ASAP w/o MAML | 3.97% | 3.01% | **0.96%** | 0.5m | 10.00% | 8.11% | 1.89% | 5.3m | 5.11% | 3.78% | 1.33% | 1.1m | **12.08%** | 9.51% | 2.57% | 10.7m |
| INViT-1V+MAML | 3.75% | 3.23% | 0.52% | 0.4m | 11.27% | 9.75% | 1.52% | 4.5m | 5.75% | 4.63% | 1.12% | 1.0m | 12.43% | 10.69% | 1.74% | 10.3m |
| INViT-1V+ASAP w/ MAML | **3.53%** | **2.69%** | 0.84% | 0.5m | **9.67%** | **6.99%** | **2.68%** | 5.3m | **4.92%** | **3.53%** | 1.39% | 1.1m | 12.23% | **9.49%** | **2.74%** | 10.7m |
| INViT-1V+ASAP w/ MAML(Tune All) | **3.53%** | 3.01% | 0.52% | 0.5m | **9.67%** | 7.53% | 2.14% | 5.3m | **4.92%** | 4.03% | 0.89% | 1.1m | 12.23% | 11.02% | 1.21% | 10.7m |
| **Explosion** | | | | | | | | | | | | | | | | |
| (Near-)Optimality | 0.00% | / | / | 2.9m | 0.00% | / | / | 17.5h | 0.00% | / | / | 30.0m | 0.00% | / | / | 2.8d |
| POMO | 2.87% | 2.86% | 0.01% | 0.3m | 54.24% | 54.11% | 0.13% | 2.3m | 7.11% | 7.07% | 0.04% | 0.5m | 25.19% | 24.47% | 0.72% | 3.7m |
| POMO+ASAP w/o MAML | 2.73% | 2.61% | 0.12% | 0.3m | 52.15% | 46.17% | 5.98% | 2.3m | **6.22%** | 4.55% | 1.67% | 0.5m | 20.22% | 15.83% | 4.39% | 3.7m |
| POMO+MAML | 2.69% | 2.63% | 0.06% | 0.3m | 51.18% | 48.23% | 2.95% | 2.3m | 6.32% | 5.11% | 1.21% | 0.5m | 23.12% | 21.06% | 2.06% | 3.7m |
| POMO+ASAP w/ MAML | **2.65%** | 2.47% | 0.18% | 0.3m | 50.35% | 44.26% | **6.09%** | 2.3m | 6.25% | **4.43%** | **1.82%** | 0.5m | **19.78%** | **14.33%** | **5.45%** | 3.7m |
| POMO+ASAP w/ MAML(Tune All) | **2.65%** | **2.43%** | **0.22%** | 0.3m | 50.35% | 49.30% | 1.05% | 2.3m | 6.25% | 4.92% | 1.33% | 0.5m | **19.78%** | 16.75% | 3.03% | 3.7m |
| INViT-1V | 6.33% | 5.80% | 0.53% | 0.4m | 15.06% | 13.72% | 1.34% | 4.5m | 6.23% | 6.26% | -0.03% | 1.0m | 14.18% | 13.63% | 0.55% | 10.3m |
| INViT-1V+ASAP w/o MAML | 3.57% | 2.85% | 0.72% | 0.5m | 11.21% | 9.51% | 1.70% | 5.3m | **5.19%** | 4.17% | 1.02% | 1.1m | 12.66% | 9.67% | 2.99% | 10.8m |
| INViT-1V+MAML | 3.79% | 3.21% | 0.58% | 0.4m | 14.17% | 12.41% | 1.76% | 4.5m | 5.88% | 5.05% | 0.83% | 1.0m | 13.55% | 11.29% | 2.26% | 10.3m |
| INViT-1V+ASAP w/ MAML | **3.37%** | **2.31%** | **1.06%** | 0.5m | **10.35%** | 8.27% | **2.08%** | 5.3m | 5.29% | 4.15% | 1.14% | 1.1m | **12.19%** | **8.95%** | **3.24%** | 10.8m |
| INViT-1V+ASAP w/ MAML(Tune All) | **3.37%** | 2.91% | 0.46% | 0.5m | **10.35%** | 9.33% | 1.02% | 5.3m | 5.29% | **4.00%** | **1.29%** | 1.1m | **12.19%** | 9.92% | 2.27% | 10.8m |

INViT-1V baselines versus their counterparts enhanced with ASAP. For **generalization performance** (results "Before" adaptation), incorporating ASAP consistently reduces the optimality gap. On the Clustered TSP-100 dataset, POMO+ASAP w/o MAML reduces the gap to 4.32% from POMO's 5.22%, and INViT-1V+ASAP w/o MAML improves the gap to 4.80% compared to 7.14% for the standard INViT-1V. In terms of **adaptation performance**, ASAP proves critical for preventing performance degradation and boosting convergence. In the Clustered TSP-1000 task, while the standard POMO policy suffers a negative improvement of -10.98% after adaptation, POMO+ASAP w/o MAML achieves a positive improvement of 7.87%. Similarly, on the Implosion CVRP-100 dataset, while standard INViT-1V stagnates with a -0.01% change, INViT-1V+ASAP w/o MAML yields a robust 2.57% improvement, highlighting the method's ability to guide the model toward better solutions during online search. See more results on TSPLIB/CVRPLIB in Section F.2.1.

**Time Cost Analysis.** We analyze the inference time overhead introduced by ASAP. As observed in Table 1, the time cost increase is negligible relative to the performance gains. For example, on the ID-Large dataset, the inference time for PCT+ASAP w/o MAML is 1.1 minutes compared to 1.0 minutes for the baseline PCT, representing a marginal

increase. Similarly, in the TSP/CVRP experiments (Table 2), INViT-1V+ASAP w/o MAML requires 0.5 minutes for Clustered TSP-100 compared to 0.4 minutes for the baseline. Even in computationally heavier tasks like CVRP-100 (Evaluation on CVRP-500), the time difference between POMO (3.7m) and POMO+ASAP w/o MAML (3.7m) is virtually non-existent, suggesting that ASAP is computationally efficient and suitable for practical deployment. For training and adaptation time evaluation, refer to Section F.1.3 and Section F.2.3.

**Ablation Study.** To isolate the contributions of different components, we conducted an ablation study analyzing the impact of MAML and ASAP. First, comparing w/o MAML vs. w/ MAML: Applying MAML generally improves the base model's initialization. For instance, in Table 1 (ID-Large), PCT+MAML (71.7%) outperforms the standard PCT (70.0%). In Table 2, POMO+MAML (5.07%) also shows a lower initial gap than POMO (5.22%) on Clustered TSP-100. Second, comparing w/o ASAP vs. w/ ASAP (after applying MAML): The addition of ASAP yields further significant gains on top of MAML. In Table 1, PCT+ASAP w/ MAML achieves 73.5% utilization on ID-Large, clearly surpassing PCT+MAML at 71.7%. Furthermore, the adaptation capability is enhanced; PCT+ASAP w/ MAML shows a 0.7% improvement after adaptation compared to only

0.3% for PCT+MAML. This pattern holds in Table 2, where POMO+ASAP w/ MAML achieves a 4.15% gap on Clustered TSP-100, significantly better than the 5.07% gap of POMO+MAML, demonstrating that ASAP provides distinct benefits beyond standard meta-learning initializations. Third, comparing selective adaptation vs. full-parameter tuning (ASAP w/ MAML vs. ASAP w/ MAML (Tune All)): The results indicate that updating all parameters during online adaptation diminishes performance gains, likely due to overfitting on limited online instances. For example, in Table 1 (ID-Large), while PCT+ASAP w/ MAML(Tune All) matches the pre-adaptation utilization of 73.5%, its post-adaptation improvement drops to +0.2%, well below the +0.7% achieved by our targeted adaptation approach. This trend is strongly mirrored in Table 2; on Clustered TSP-100, POMO+ASAP w/ MAML(Tune All) produces a mere 0.10% optimality gap improvement compared to the 0.33% improvement of POMO+ASAP w/ MAML, resulting in a worse final post-adaptation gap (4.05% vs. 3.82%). For sensitivity analysis, refer to Section F.1.2 and Section F.2.2.

## 6. Conclusion

In this work, we introduce Adaptive Selection After Proposal (ASAP), a generic framework designed to address the critical challenge of distribution shift in Neural Combinatorial Optimization. By identifying the Satisficing Generalization Edge and decoupling the decision-making process into a robust proposal policy and a lightweight selection policy, our approach effectively balances cross-distribution generalization with rapid online adaptation. This architecture enables the model to freeze the generalizable proposal parameters while quickly fine-tuning the selection policy on a significantly reduced action space, ensuring efficiency during inference. Comprehensive experiments across the 3D Bin Packing Problem (3D-BPP), TSP, and CVRP demonstrate that ASAP consistently outperforms state-of-the-art baselines in both zero-shot generalization and few-shot adaptation scenarios. Ultimately, our findings confirm that explicitly separating candidate generation from final decision-making is a pivotal strategy for developing robust, scalable, and adaptable neural solvers for complex combinatorial problems.

## Acknowledgments

This work has been supported in part by the program of National Natural Science Foundation of China (No. 62176154), the program of National Natural Science Foundation of China (No. 62503322), the AI for Science Seed Program of Shanghai Jiao Tong University (project number 2025AI4SQY06), the Shanghai Jiao Tong University Xiaomi Scholar Fund, and the Shanghai Municipal Special Program for Basic Research on General AI Foundation Mod-

els.

## Impact Statement

This paper presents work aimed at advancing Neural Combinatorial Optimization, specifically for logistics-centric problems such as 3D Bin Packing and Vehicle Routing. The primary societal consequence of this research is the potential for increased efficiency in global supply chains and industrial operations. By improving the generalization capability and online adaptation of neural solvers, this method contributes to higher space utilization in shipping containers and more optimal route planning. While the automation of optimization tasks carries general implications for workforce roles in logistics planning, the proposed framework aims to create more robust and reliable tools that can handle dynamic, real-world distribution shifts without failure.

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

# A. Detailed Problem Formulation

## A.1. 3D-BPP

An *online regular 3D-BPP* instance contains two elements: a container of size $(L, W, H)$ and a sequence $(l_i, w_i, h_i)_{i=1}^n$ of $n$ cuboid-shaped items of size $(l_i, w_i, h_i)$. In this problem, the solver needs to place each item in this sequence without knowing any information about the subsequent items. Denote the placement of each item as $p_i = (x_i, y_i, z_i)$, the following constraints are the solver has to follow: (1) *non-overlapping constraint* (i.e., placed items cannot intersect in the 3D space), which has the form:

$$\begin{cases} x_i + l_i \leq x_j + L(1 - e_{ij}^x) \\ y_i + w_i \leq y_j + W(1 - e_{ij}^y) \\ z_i + h_i \leq z_j + H(1 - e_{ij}^z) \end{cases}$$

where $e_{ij}^x, e_{ij}^y, e_{ij}^z$ takes value 1 otherwise 0 if item $i$ precedes item $j$ along $x, y, z$ axis. (2) *containment constraint* (i.e., placed items should fit inside the container), which has the form:

$$\begin{cases} 0 \leq x_i \leq L - l_i \\ 0 \leq y_i \leq W - w_i \\ 0 \leq z_i \leq H - h_i \end{cases}$$

Once placed, an item cannot be moved. The objective is to place a maximum number of items in order to maximize the space utilization of the container, which represents the proportion of the volume used in the container. As such, it is upperbounded by 1. If the first $T$ items fit in the container, it is defined as follows:

$$\text{Uti} = \sum_{i=1}^{T} \frac{l_i w_i h_i}{LWH}. \tag{1}$$

We assume that the items in an online 3D-BPP instance are sampled from some probability distribution, which may change during test.

## A.2. TSP

A *Euclidean TSP* instance contains a sequence of $n$ cities (nodes) characterized by their coordinates $c_i = (x_i, y_i)$ for $i = 1, \ldots, n$. In this problem, the solver needs to find a permutation $\pi$ of the set $\{1, \ldots, n\}$ representing the order in which the cities are visited. Denote the tour as a sequence of indices $(\pi(1), \pi(2), \ldots, \pi(n))$, the solver must satisfy the following condition: (1) *cycle constraint* (i.e., the tour must visit every city exactly once and return to the start), which implies that $\pi$ must be a bijection from $\{1, \ldots, n\}$ to itself.

The objective is to minimize the total travel distance of the tour. If we denote $d(c_i, c_j) = \|c_i - c_j\|_2$ as the Euclidean distance between two cities, the total tour length is defined as follows:

$$\text{Cost}_{\text{TSP}} = \sum_{i=1}^{n-1} d(c_{\pi(i)}, c_{\pi(i+1)}) + d(c_{\pi(n)}, c_{\pi(1)}). \tag{2}$$

Similar to the BPP formulation, we typically assume that the coordinates in a TSP instance are sampled from a uniform distribution (e.g., in the unit square $[0, 1]^2$).

## A.3. CVRP

A *Capacitated Vehicle Routing Problem (CVRP)* instance contains a depot node with coordinates $c_0 = (x_0, y_0)$, a sequence of $n$ customer nodes characterized by coordinates $c_i = (x_i, y_i)$ and demands $q_i$ for $i = 1, \ldots, n$, and a vehicle capacity $Q$. The solver needs to partition the customers into $K$ routes (sub-tours), denoted as $R_1, \ldots, R_K$. Each route $R_k$ is a sequence of customers starting and ending at the depot $c_0$. The following constraints are the solver has to follow: (1) *capacity constraint* (i.e., the total demand of customers in a single vehicle cannot exceed its capacity), which has the form:

$$\sum_{i \in R_k} q_i \leq Q, \quad \forall k \in \{1, \ldots, K\}$$

(2) *visiting constraint* (i.e., each customer must be served exactly once), meaning that the union of all customers in routes $R_1, \ldots, R_K$ is exactly $\{1, \ldots, n\}$ and the intersection of any two routes (excluding the depot) is empty.

The objective is to minimize the total distance traveled by all vehicles. If $L(R_k)$ denotes the length of route $R_k$ (including travel from and to the depot), the total cost is defined as follows:

$$\text{Cost}_{\text{CVRP}} = \sum_{k=1}^{K} L(R_k).$$ (3)

As with previous formulations, the customer locations and demands are often assumed to be sampled from specific probability distributions.

## B. Theoretical Analysis of ASAP

In this part, we analyze the behavior of ASAP within a single timestep to validate the conclusion presented in Section Section 4.1. We first formalize the decision-making process of ASAP mathematically: Let $\mathcal{A}$ be the finite set of feasible placement actions available at a given timestep $t$. Let $n = |\mathcal{A}|$ denote the total number of available actions. **Let $\mathbb{T} \subseteq \mathcal{A}$ be the set of optimal actions (which we also term it as target actions in our following discussion), defined as those that maximize the Q-value function $Q_{optim}(s, a)$.** Let $\pi_\theta$ be a policy parameterized by weights $\theta$. The policy outputs a probability distribution over $\mathcal{A}$, assigning a probability $p_i$ to each action $a_i \in \mathcal{A}$, such that $\sum_{i=1}^{n} p_i = 1$. Specifically, let $p_{t_1}$ denote the probability assigned to the target action $t_1$ by the policy.

In our setup, we adopt a two-stage process: we first sample $k$ actions from the original action set, then select the final action from these $k$ sampled actions. In practice, the two-stage process employs two distinct policies: the proposal policy is fixed, while the selection policy undergoes rapid fine-tuning. To demonstrate the superiority of our method over directly selecting actions from the original action set, we first assume the same policy is used for both the proposal and selection stages. Note that in practical implementation, since the selection policy is designed for fine-tuning, it will yield superior performance compared to using the same policy for both stages.

To calculate the probability, we first introduce a simplified assumption: there exists only one target action, i.e., $|\mathbb{T}| = 1$. Let the index of this target action be $t_1$, and let $p_{t_1}$ denote the probability assigned to $t_1$ by the policy. Under this assumption, the probability of selecting the target action via one-stage decision process is straightforward: it equals the policy-assigned probability of the target action, i.e., $p_{t_1}$. For the two-stage decision process, we first analyze the proposal stage (the first stage, where $k$ actions are sampled from the original $n$ actions). Notably, sampling $k$ candidate actions simultaneously is equivalent to sequential sampling without replacement (i.e., no action is sampled more than once). We thus decompose the proposal stage step-by-step as follows:

- In the first sampling step, the only way to select the target action is to directly sample $t_1$. Since no actions have been excluded yet, the probability is exactly the policy-assigned probability of $t_1$: $\mathbb{P}_1^{\text{pro}} = p_{t_1}$

- To select the target action in the second step, two conditions must be satisfied: The first sampled action is a non-target action (i.e., $a_j$ where $j \neq t_1$); The second sampled action is the target action $t_1$. The probability $\mathbb{P}_2^{\text{pro}}$ is the sum of joint probabilities for all valid $(a_j, t_1)$ pairs. For each non-target action $a_j$, the joint probability of sampling $a_j$ first and $t_1$ second is $\frac{p_j p_{t_1}}{1 - p_j}$—the denominator $1 - p_j$ accounts for excluding $a_j$ after the first sampling. Summing over all non-target $j$ gives: Following this logic, we can have

$$\mathbb{P}_2^{\text{pro}} = \sum_{j \neq t_1} \frac{p_j p_{t_1}}{1 - p_j}$$
$$= p_{t_1} \sum_{j \neq t_1} \frac{p_j}{1 - p_j}$$
$$= p_{t_1} \cdot (n-1) \sum_{j \neq t_1} \frac{1}{n-1} \cdot \frac{p_j}{1 - p_j}$$

We apply Jensen's inequality here: since $f(x) = \frac{x}{1-x}$ is a convex function on $[0, 1)$, the weighted average of $f(x)$ is at least $f$ of the weighted average. The weights are $\frac{1}{n-1}$ (one for each non-target action), and the weighted average of $p_j$

(over $j \neq t_1$) is $\frac{1-p_{t_1}}{n-1}$ (since $\sum_{j \neq t_1} p_j = 1 - p_{t_1}$). Substituting this into the inequality:

$$\mathbb{P}_2^{\text{pro}} \geq p_{t_1} \cdot (n-1) \cdot \frac{\frac{1-p_{t_1}}{n-1}}{1 - \frac{1-p_{t_1}}{n-1}}$$

$$\geq p_{t_1} \cdot (n-1) \cdot \frac{1-p_{t_1}}{n-1-(1-p_{t_1})}$$

$$\geq p_{t_1} \cdot (n-1) \cdot \frac{1-p_{t_1}}{n-1}$$

$$\geq p_{t_1} \cdot (1-p_{t_1})$$

- For the $m$-th sampling step ($3 \leq m \leq k$), the target action is selected only if the first $m-1$ steps all sample non-target actions, and the $m$-th step samples $t_1$. Mathematically, this is expressed by summing over all possible sequences of $m-1$ non-target actions $(s_1, ..., s_{m-1})$ (where $s_j \neq t_1$):

$$\mathbb{P}_m^{\text{pro}} = \sum_{\substack{(s_1,...,s_{m-1}) \\ s_j \in \{1,...,n\} \setminus \{t_1\}}} \left( \prod_{j \in \{1,...,m-1\}} \frac{p_{s_j}}{1 - \sum_{d<j} p_{s_d}} \right) \cdot \frac{p_{t_1}}{1 - \sum_{j \in \{1,...,m-1\}} p_{s_j}}$$

The product term $\prod_{j=1}^{m-1} \frac{p_{s_j}}{1-\sum_{d<j} p_{s_d}}$ is the probability of sampling the sequence $(s_1, ..., s_{m-1})$ (denominators exclude previously sampled actions). The final term $\frac{p_{t_1}}{1-\sum_{j=1}^{m-1} p_{s_j}}$ is the probability of sampling $t_1$ in the $m$-th step (after excluding the first $m-1$ actions). To simplify, let $A = 1 - \sum_{j=1}^{m-2} p_{s_j}$ (the remaining probability mass after $m-2$ non-target samples). The denominator $1 - \sum_{j=1}^{m-1} p_{s_j}$ becomes $A - p_{s_{m-1}}$, so:

$$\mathbb{P}_m^{\text{pro}} = p_{t_1} \cdot \sum_{(s_1,...,s_{m-2})} \left( \prod_{j=1}^{m-2} \frac{p_{s_j}}{1 - \sum_{d<j+1} p_{s_d}} \right) \cdot \sum_{s_{m-1}} \frac{p_{s_{m-1}}}{A - p_{s_{m-1}}}.$$

Applying Jensen's inequality again (using convexity of $f(x) = \frac{x}{A-x}$), let $N = n - m$ (number of remaining non-target actions after $m-2$ samples) and $M = A - p_{t_1}$ (total probability of remaining non-target actions). The inner sum satisfies:

$$\sum_{s_{m-1}} \frac{p_{s_{m-1}}}{A - p_{s_{m-1}}} \geq N \cdot \frac{\frac{M}{N}}{A - \frac{M}{N}} = \frac{A - p_{t_1}}{A - \frac{1}{n-m}(A - p_{t_1})}.$$

Substituting back, we link $\mathbb{P}_m^{\text{pro}}$ to $\mathbb{P}_{m-1}^{\text{pro}}$ (the probability of selecting non-target actions in the first $m-2$ steps):

$$\mathbb{P}_m^{\text{pro}} \geq \mathbb{P}_{m-1}^{\text{pro}} \cdot \frac{A - p_{t_1}}{A - \frac{1}{n-m}(A - p_{t_1})}.$$

We further prove $\frac{A-p_{t_1}}{A - \frac{1}{n-m}(A-p_{t_1})} > 1 - p_{t_1}$ by algebraic rearrangement: Cross-multiply (denominators are positive, so inequality direction remains):

$$(A - p_{t_1}) > (1 - p_{t_1}) \cdot \left( A - \frac{1}{n-m}(A - p_{t_1}) \right).$$

Expand and simplify to a quadratic in $p_{t_1}$:

$$p_{t_1}^2 - [(A+1) + (n-m) - A(n-m)] p_{t_1} + A(n-m) > 0.$$

To verify this inequality, we compute the discriminant $\Delta$ of the quadratic equation $a p_{t_1}^2 + b p_{t_1} + c = 0$ with coefficients:

$$a = 1, \quad b = -[(A+1) + (n-m)(1-A)], \quad c = A(n-m).$$

The discriminant is given by:

$$\Delta = b^2 - 4ac$$
$$= [(A+1) + (n-m)(1-A)]^2 - 4A(n-m).$$

Since $A$ represents the remaining probability mass, we typically have $A \approx 1$. In the limit where $A \to 1$, the term $(n-m)(1-A)$ approaches 0, simplifying the discriminant to:

$$\Delta \approx (1+1)^2 - 4(1)(n-m) = 4 - 4(n-m).$$

Since $m \ll n$, we have $n - m \geq 1$, which implies $\Delta < 0$. As the discriminant is negative and the leading coefficient ($a = 1$) is positive, the quadratic is always positive, proving the inequality.

Finally, substituting back gives: $\mathbb{P}_m^{\text{pro}} \geq \mathbb{P}_{m-1}^{\text{pro}}(1 - p_{t_1})$.

- Summing the probabilities of selecting $t_1$ in any of the $k$ sampling steps, we use the recursive relation $\mathbb{P}_m^{\text{pro}} \geq \mathbb{P}_{m-1}^{\text{pro}}(1 - p_{t_1})$:

$$\mathbb{P}^{\text{pro}} = \sum_{t=1}^{k} \mathbb{P}_t^{\text{pro}}$$
$$\geq p_{t_1} + (1 - p_{t_1})p_{t_1} + (1 - p_{t_1})^2 p_{t_1} + ... + (1 - p_{t_1})^{k-1} p_{t_1}.$$

This is a geometric series with first term $a = p_{t_1}$ and common ratio $r = 1 - p_{t_1}$. Using the geometric series sum formula $\sum_{i=0}^{k-1} ar^i = a \cdot \frac{1-r^k}{1-r}$:

$$\mathbb{P}^{\text{pro}} \geq p_{t_1} \cdot \frac{1 - (1 - p_{t_1})^k}{1 - (1 - p_{t_1})} = 1 - (1 - p_{t_1})^k.$$

For the selection process, we can calculate the expectation of the probability for the selected actions by

$$E[\mathbb{P}^{total}] = p_{t_1} + \frac{k-1}{n-1} \cdot (1 - p_{t_1})$$

Therefore, we can get the selection probability by

$$\mathbb{P}^{\text{sel}} = \frac{p_{t_1}}{E[p^{total}]} = \frac{p_{t_1}}{p_{t_1} + \frac{k-1}{n-1} \cdot (1 - p_{t_1})}$$

**Theorem B.1** (Superiority of Two-Stage Decision Process). *Given a policy $\pi$ assigning probability $p_{t_1}$ to the optimal action, the probability of selecting the optimal action via the Two-Stage process ($\mathbb{P}^{two}$) strictly exceeds the probability via the One-Stage process ($\mathbb{P}^{one}$), i.e., $\mathbb{P}^{two} > \mathbb{P}^{one}$, if:*

$$p_{t_1} > \frac{1}{n-1}.$$

First, we can get the two-stage probability by

$$\mathbb{P}^{two} = \mathbb{P}^{\text{pro}} \cdot \mathbb{P}^{\text{sel}} \geq (1 - (1 - p_{t_1})^k) \cdot \frac{p_{t_1}}{p_{t_1} + \frac{k-1}{n-1} \cdot (1 - p_{t_1})}$$

Compared to one-stage selection probability $p_i$, we want to analyze the condition that

$$(1 - (1 - p_{t_1})^k) \cdot \frac{p_{t_1}}{p_{t_1} + \frac{k-1}{n-1} \cdot (1 - p_{t_1})} \geq p_{t_1}$$

We can transfer this equation to

$$(1 - (1 - p_{t_1})^k) \cdot \frac{p_{t_1}}{p_{t_1} + \frac{k-1}{n-1} \cdot (1 - p_{t_1})} \geq p_{t_1}$$

$$1 - (1 - p_{t_1})^k \geq p_{t_1} + \frac{k-1}{n-1} \cdot (1 - p_{t_1})$$

$$(1 - p_{t_1}) - (1 - p_{t_1})^k \geq \frac{k-1}{n-1} \cdot (1 - p_{t_1})$$

$$1 - (1 - p_{t_1})^{k-1} \geq \frac{k-1}{n-1}$$

$$(1 - p_{t_1})^{k-1} \leq \frac{n-k}{n-1}$$

$$(k-1) \ln(1 - p_{t_1}) \leq \ln\left(\frac{n-k}{n-1}\right)$$

$$\ln(1 - p_{t_1}) \leq \frac{1}{k-1} \ln\left(\frac{n-k}{n-1}\right)$$

$$1 - p_{t_1} \leq \left(\frac{n-k}{n-1}\right)^{\frac{1}{k-1}}$$

$$p_{t_1} \geq 1 - \left(\frac{n-k}{n-1}\right)^{\frac{1}{k-1}}$$

In our two-stage selection process, we can have that $n >> k$ since we only propose a small action set, we can have

$$\left(\frac{n-k}{n-1}\right)^{\frac{1}{k-1}} = \left(1 - \frac{k-1}{n-1}\right)^{\frac{k-1}{n-1} \cdot \frac{1}{n-1}} \approx e^{-\frac{1}{n-1}} \approx 1 - \frac{1}{n-1}.$$

which means that when

$$p_{t_1} \geq 1 - (1 - \frac{1}{n-1}) = \frac{1}{n-1},$$

the two-stage selection probability is higher than one-stage selection. This condition is easy to satisfy since we use a well-trained policy, which is likely to assign a higher probability for the valuable action though it might not assign the highest probability.

**For the situation when the number of the target actions is greater than 1, it is obvious that the two-stage probability is higher than one-stage since we can regard selecting each target action as independent events.**

**Theorem B.2** (Robustness of Proposal Set Inclusion). *Let $\pi$ be a policy assigning probability $p_{t_1}$ to the optimal action $a_{t_1}$, and let $\pi^*$ be the optimal policy assigning probability $p_{t_1}^*$ to the same action. If the proposal set size $k$ exceeds a threshold $k_0$ given by:*

$$k > k_0 \approx \frac{\ln\left((n-1)p_{t_1}^* p_{t_1}\right)}{p_{t_1}}$$

*then we have:*

$$\mathbb{P}^{pro} > \mathbb{P}^{sel}, \quad \frac{\partial \mathbb{P}^{pro}}{\partial p_{t_1}} < \frac{\partial \mathbb{P}^{sel}}{\partial p_{t_1}}$$

In this part, we start from assuming the number of target actions is 1. First, we analyze the sensitivity condition $\frac{\partial \mathbb{P}^{pro}}{\partial p_{t_1}} < \frac{\partial \mathbb{P}^{sel}}{\partial p_{t_1}}$. As we mentioned before, the optimal gap happens since the mismatch between policy-induced probability and the optimal probability. For the case that the policy-induced probability gives a higher probability for the target action than the optimal probability, it is fine because we wish the policy select the target action. Therefore, we only need to consider the case that the target action is overlooked by the policy. We need to prove that in this situation, the influence on the proposal process is less than that on the selection process. We suppose given a specific distribution of incoming items, the optimal probability of the target action is $p_{t_1}^*$. Then we can get that the difference between the optimal proposal probability satisfy

$$\Delta \mathbb{P}^{pro} \leq \left(1 - (1 - p_{t_1}^*)^k)\right) - \left(1 - (1 - p_{t_1})^k)\right)$$
$$= (1 - p_{t_1})^k - (1 - p_{t_1}^*)^k$$

and the difference between the optimal selection probability can be derived as

$$\Delta \mathbb{P}^{\text{sel}} = \frac{p_{t_1}^*}{p_{t_1}^* + \frac{k-1}{n-1} \cdot (1 - p_{t_1}^*)} - \frac{p_{t_1}}{p_{t_1} + \frac{k-1}{n-1} \cdot (1 - p_{t_1})}$$

$$= \frac{\frac{k-1}{n-1} \cdot (p_{t_1}^* - p_{t_1})}{\left(p_{t_1}^* + \frac{k-1}{n-1} \cdot (1 - p_{t_1}^*)\right)\left(p_{t_1}^* + \frac{k-1}{n-1} \cdot (1 - p_{t_1}^*)\right)}$$

Since $n >> k > 1$, $\frac{k-1}{n-1} << 1$, we can estimate that $\frac{k-1}{n-1}(1-p) << p$, therefore, we can have

$$\Delta \mathbb{P}^{\text{sel}} \approx \frac{(k-1) \cdot (p_{t_1}^* - p_{t_1})}{(n-1) \cdot p_{t_1}^* p_{t_1}}$$

Since we can find that $\Delta \mathbb{P}^{\text{pro}}$ is monotonically decreasing when $k$ is increasing and $\Delta \mathbb{P}^{\text{sel}}$ is monotonically increasing, we can calculate $k_0$ such that when $k_0 < k << n$, $\Delta \mathbb{P}^{\text{pro}} < \Delta \mathbb{P}^{\text{sel}}$ as shown below.

$$\frac{(k_0 - 1) \cdot (p_{t_1}^* - p_{t_1})}{(n-1) \cdot p_{t_1}^* p_{t_1}} = (1 - p_{t_1})^k - (1 - p_{t_1}^*)^k$$

$$\frac{(k_0 - 1) \cdot (p_{t_1}^* - p_{t_1})}{(n-1) \cdot p_{t_1}^* p_{t_1}} = (1 - p_{t_1})^k - (1 - p_{t_1}^*)^k \approx e^{-k_0 p_{t_1}} - e^{-k_0 p_{t_1}^*}$$

$$\frac{(k_0 - 1) \cdot (p_{t_1}^* - p_{t_1})}{(n-1) \cdot p_{t_1}^* p_{t_1}} \approx e^{-k_0 p_{t_1}} - e^{-k_0 p_{t_1}^*} \approx e^{-k_0 p_{t_1}} - e^{-k_0 p_{t_1}}(1 - k_0 \Delta p)$$

$$\frac{(k_0 - 1) \cdot (p_{t_1}^* - p_{t_1})}{(n-1) \cdot p_{t_1}^* p_{t_1}} \approx e^{-k_0 p_{t_1}} k_0 \Delta p$$

$$k_0 \approx \frac{\ln \left((n-1) p_{t_1}^* p_{t_1}\right)}{p_{t_1}}$$

Therefore, we can say that when $k > k_0 \approx \frac{\ln\left((n-1) p_{o_1} p_{t_1}\right)}{p_{t_1}}$, the proposal policy is easier to generalize than the selection policy. In real implementation, this value is usually acceptable, i.e., suppose $p_{t_1}^* = 0.5, p_{t_1} = 0.3, n = 50$, then $k_0 \approx 5$.

Next, we prove the condition $\mathbb{P}^{pro} > \mathbb{P}^{sel}$. Substituting the derived formulas, we require:

$$1 - (1 - p_{t_1})^k > \frac{p_{t_1}}{p_{t_1} + \frac{k-1}{n-1}(1 - p_{t_1})}.$$

Since $p_{t_1} < 1$, we can approximate $(1 - p_{t_1})^k \approx 1 - k p_{t_1}$ for small $p_{t_1}$. Thus $\mathbb{P}^{pro} \approx k p_{t_1}$. For the selection probability, if $n$ is large, let $\epsilon = \frac{k-1}{n-1}$. Then $\mathbb{P}^{sel} \approx \frac{p_{t_1}}{p_{t_1}+\epsilon}$. The condition $\mathbb{P}^{pro} > \mathbb{P}^{sel}$ approximates to $k p_{t_1} > \frac{p_{t_1}}{p_{t_1}+\epsilon}$, which implies $k(p_{t_1} + \epsilon) > 1$. Since $\epsilon > 0$, a sufficient condition for this inequality is $k p_{t_1} > 1$, or equivalently $k > \frac{1}{p_{t_1}}$. We compare this with the condition derived from sensitivity analysis, $k > k_0$.

$$k_0 \approx \frac{\ln \left((n-1) p_{t_1}^* p_{t_1}\right)}{p_{t_1}}.$$

In typical large action spaces where $n \gg 1$, the term inside the logarithm $C = (n-1) p_{t_1}^* p_{t_1}$ is generally greater than $e$ (Euler's number). Consequently, $\ln(C) > 1$. This implies:

$$k_0 = \frac{\ln(C)}{p_{t_1}} > \frac{1}{p_{t_1}}.$$

Therefore, satisfying the sensitivity condition $k > k_0$ automatically satisfies $k > \frac{1}{p_{t_1}}$, which in turn ensures $k(p_{t_1} + \epsilon) > 1$. Thus, when $k > k_0$, both the robustness condition and the probability superiority condition $\mathbb{P}^{pro} > \mathbb{P}^{sel}$ are met. **For the case that the number of target actions is greater than 1, we can still follow the same logic since we can regard selecting each target action as independent events.**

# C. Detailed Preliminary Experiments

To demonstrate that our proposed two-stage design is grounded in both theory and practice, we empirically validate the theorem presented in B through preliminary experiments.

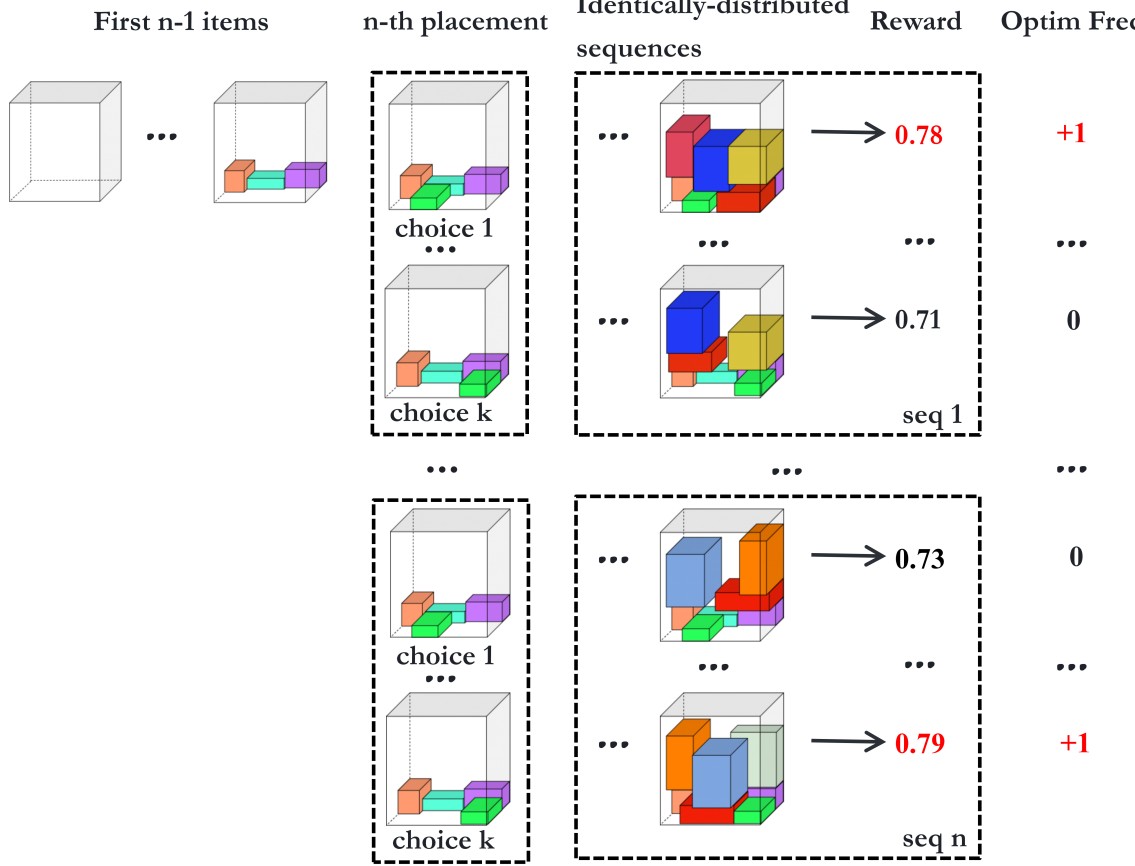

*Figure 4.* Demonstration of MCTS experiments.

### C.1. Experimental Setup

Figure 4 illustrates the methodology used to approximate an optimal policy via Monte Carlo Tree Search (MCTS) on 3D-BPP. Given the state of the first $n - 1$ items, we evaluate potential candidates for the $n$-th placement. To capture the stochastic nature of the problem, we generate a set of identically distributed sequences of future items drawn from the source distribution. For each generated sequence, we expand every available candidate node (Choice 1 through $k$) and simulate the subsequent placement steps using a pre-trained DRL policy to determine the final packing configuration. We then identify which specific choice yields the maximum reward for that sequence. By aggregating the results across all generated sequences, we compute the "Optimization Frequency"—the probability that a specific node leads to the best possible outcome. As the number of sampled sequences increases, this frequency converges to an approximation of the optimal policy distribution. Consequently, we utilize this approximated optimal frequency as a ground truth benchmark to assess the action distribution induced by the generic DRL policy, thereby isolating the factors contributing to performance degradation on novel instance distributions.

### C.2. MCTS Results

**Distribution Mismatch.** Figures 5 and 6 present detailed preliminary results across both discrete and continuous environments for the datasets defined in Section 5.1. On the Default (training) dataset, the policy-induced action frequencies align closely with the optimal frequencies, indicating that the DRL-based policy has effectively fit the training distribution. However, a divergence emerges in the Large, Medium, and Small datasets—where the item sets are subsets of the Default configuration. In these cases, the policy-induced frequency curve deviates from the optimal curve, isolating the primary

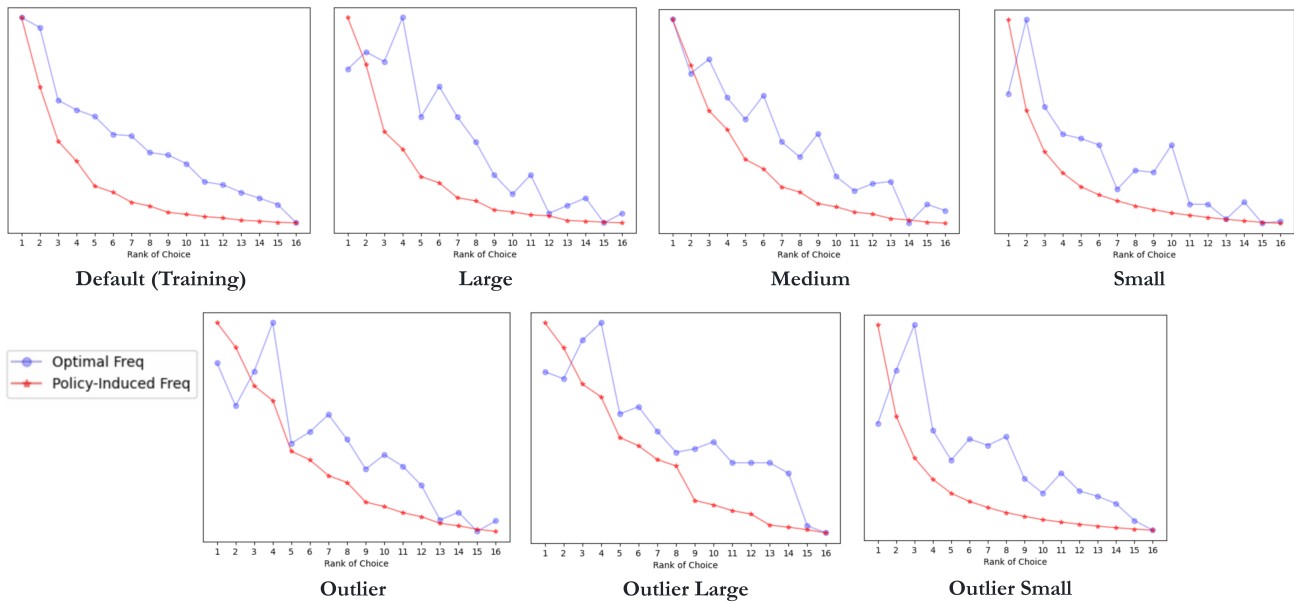

*Figure 5.* Full Preliminary Results for *Distribution Mismatch* in Discrete Environment.

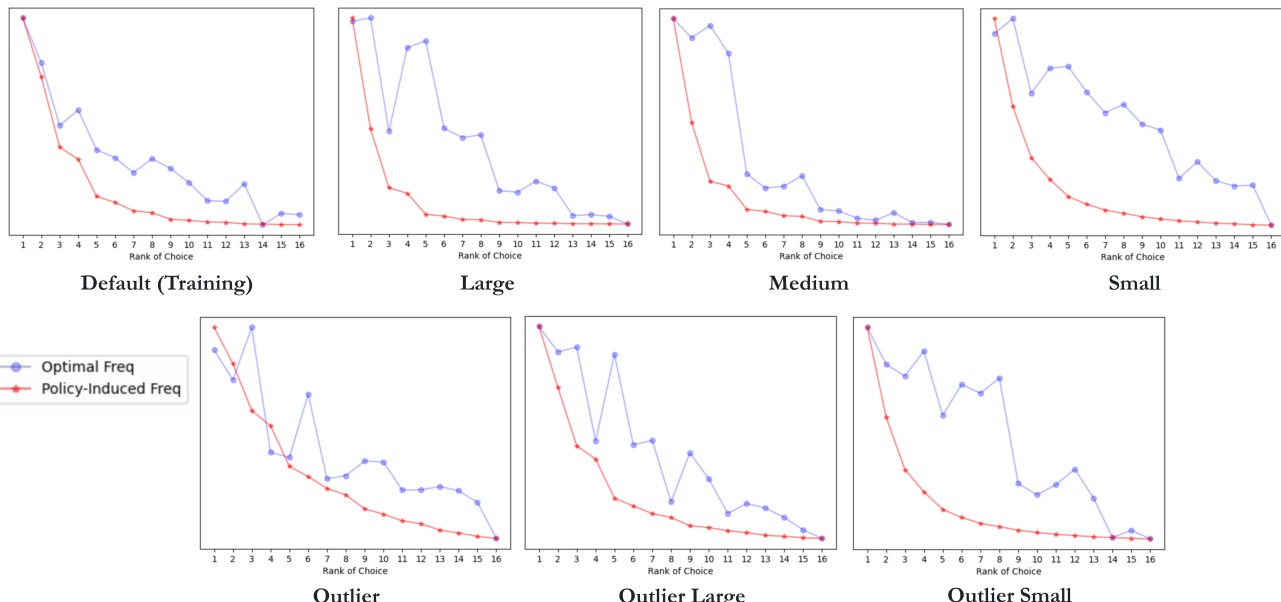

*Figure 6.* Full Preliminary Results of *Distribution Mismatch* in Continuous Environment.

cause of generalization failure: a mismatch between the decision distribution of the generic DRL policy and that of the optimal policy under the shifted instance distribution.

This phenomenon is significantly accentuated in the Out-of-Distribution (OOD) datasets (Outlier, Outlier Large, and Outlier Small), which introduce novel item sizes not seen during training. As illustrated, the deviation between the curves is severe, corroborating the analysis in Section 5.2 regarding the inferior generalizability of the DRL policy on OOD data compared to In-Distribution (ID) data. Nevertheless, despite the local mismatch on unseen distributions, a consistent global trend persists: both curves exhibit a monotonic decrease as the rank increases. Universally, actions with lower ranks correspond to lower optimal frequencies. This observation validates our two-stage design philosophy, demonstrating that *pruning* low-probability actions is a robust strategy that generalizes well across distributions, whereas precise *selection* requires adaptation to specific distributional characteristics.

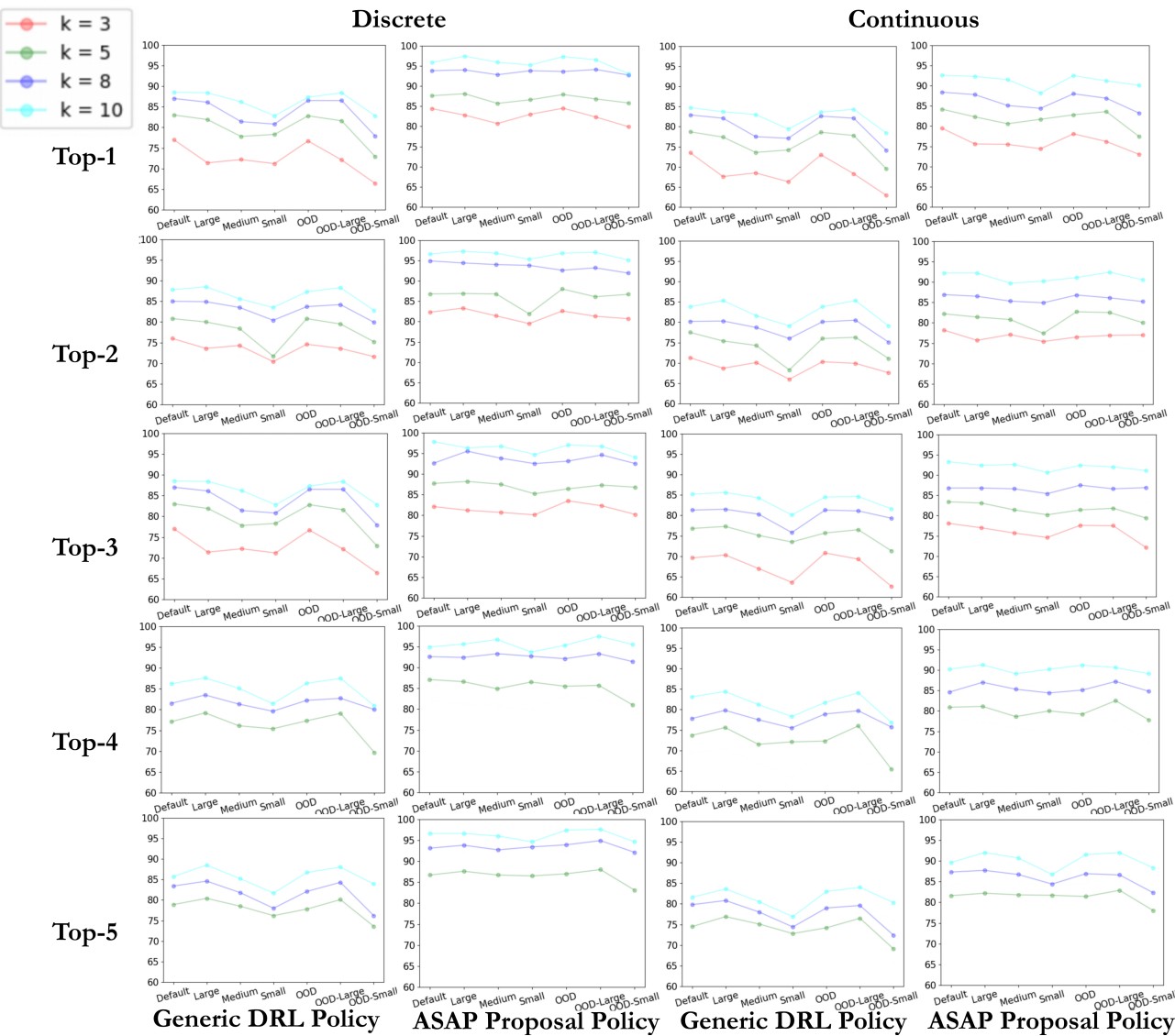

*Figure 7.* Full Preliminary Results for *Proposal Policy*.

**Proposal Policy.**    As illustrated in Figure 7, our experiments provide compelling evidence for the robustness of the proposal mechanism. While the generic DRL policy may struggle to assign the highest probability to the single optimal action (Top-1) under distributional shifts, it consistently ranks the optimal actions within the top-$k$ candidates. Detailed analysis reveals that as $k$ increases, the recall rate of the optimal action improves significantly and remains stable across both In-Distribution and Out-of-Distribution datasets. This phenomenon reinforces our core conclusion: the task of *identifying a promising subset of actions* (candidate generation) is far less sensitive to variations in item distribution than the task of *precisely predicting the optimal probability* (exact selection). Consequently, a proposal policy based on top-$k$ pruning effectively bridges the generalization gap by filtering out clearly suboptimal choices while retaining the ground truth for the subsequent refinement stage.

**Feasibility of Proposal-based MCTS.**    Complementing the preliminary findings in Section 4.1, we conducted additional experiments to validate the feasibility of using a generic DRL policy as a proposal mechanism to prune the search space. Table 3 compares the performance of an exhaustive MCTS (which explores the full action space) against our proposed method, where MCTS only expands a subset of actions suggested by the generic DRL policy. Theoretically, the computational advantage of the proposal-based approach is significant. Let $k_{full}$ denote the size of the full action space and $k_{prop}$ denote the size of the proposal set, where $k_{prop} \ll k_{full}$. For a sequence of length $n$, the worst-case search complexity is reduced

*Table 3.* MCTS performance on all datasets in discrete and continuous environments.

| | Dataset | Default | | ID-Large | | ID-Medium | | ID-Small | |
|---|---|---|---|---|---|---|---|---|---|
| | Measurement | Uti(%) | Num | Uti(%) | Num | Uti(%) | Num | Uti(%) | Num |
| Discrete | MCTS | 85.9 | 33.7 | 75.3 | 12.6 | 81.5 | 29.3 | 87.3 | 100.2 |
| | MCTS w/ Generic DRL policy proposal | 85.9 | 33.6 | 75.2 | 12.5 | 81.5 | 29.4 | 87.1 | 100.0 |
| | Dataset | OOD | | OOD-Large | | OOD-Small | | | |
| | Measurement | Uti(%) | Num | Uti(%) | Num | Uti(%) | Num | | |
| | MCTS | 68.7 | 21.6 | 65.3 | 7.4 | 84.8 | 149.5 | | |
| | MCTS w/ Generic DRL policy proposal | 68.5 | 21.5 | 65.2 | 7.4 | 84.5 | 149.1 | | |
| | Dataset | Default | | ID-Large | | ID-Medium | | ID-Small | |
| | Measurement | Uti(%) | Num | Uti(%) | Num | Uti(%) | Num | Uti(%) | Num |
| Continuous | MCTS | 66.0 | 24.1 | 65.5 | 10.8 | 67.3 | 25.2 | 70.1 | 84.0 |
| | MCTS w/ Generic DRL policy proposal | 66.0 | 23.9 | 65.4 | 10.8 | 67.2 | 25.2 | 69.8 | 83.6 |
| | Dataset | OOD | | OOD-Large | | OOD-Small | | | |
| | Measurement | Uti(%) | Num | Uti(%) | Num | Uti(%) | Num | | |
| | MCTS | 61.6 | 18.2 | 60.1 | 7.1 | 73.1 | 125.1 | | |
| | MCTS w/ Generic DRL policy proposal | 61.3 | 18.1 | 59.9 | 7.0 | 72.9 | 124.8 | | |

from $O(k_{full}^n)$ to $O(k_{prop}^n)$.

Empirically, Table 3 demonstrates that this substantial reduction in computational cost incurs negligible performance loss. Across all discrete and continuous datasets—including both In-Distribution (ID) and Out-of-Distribution (OOD) scenarios—the maximum utilization drop observed is merely $0.3\%$. In the majority of cases, the performance is identical to the exhaustive search. This result reinforces our core motivation: while a generic DRL policy may struggle to pinpoint the single optimal action on novel distributions (due to the generalization gap), it remains highly effective at identifying a small, high-quality set of candidates. Consequently, the generic policy serves as a robust filter, allowing MCTS to focus its computational resources on refining the selection among promising actions rather than exploring the entire space.

## D. MAML Implementation

---
**Algorithm 1** MAML-based Policy Training
---
**Require:** Item set $\mathcal{I}$, $\alpha, \beta$ step size hyperparameters, initialized policy $\pi_\theta$
 1: **while** not done **do**
 2:     Generate batch of distributions $p_i(\mathcal{I})$
 3:     **for** each distribution $p_i(\mathcal{I})$ **do**
 4:         Sample instance set $X_i = \{x_1, ..., x_k \sim p_i(\mathcal{I})\}$
 5:         Generate solutions $F_{X_i}(\pi_\theta) = \left\{ f_{x_j}(\pi_\theta) \right\}_{j=1}^{k}$
 6:         Compute $\theta_i' \leftarrow \theta - \alpha \nabla_\theta \mathcal{L}(F_{X_i}(\pi_\theta))$
 7:     **end for**
 8:     Update $\theta \leftarrow \theta - \beta \nabla_\theta \sum_{X_i \sim p_i(\mathcal{I})} \mathcal{L}(F_{X_i}(\pi_{\theta_i'}))$
 9: **end while**
---

The training procedure, detailed in Algorithm 1, leverages a meta-learning framework where each distinct instance distribution $p_i(\mathcal{I})$ is formulated as a unique task. At the start of each meta-iteration, we sample a batch of these distributions to simulate diverse environmental conditions. The process then proceeds in two stages: First, in the *inner loop* adaptation (lines 4-7), we sample a set of instances $X_i$ for each distribution and generate solutions using the current policy $\pi_\theta$. Using the resulting trajectories, we compute task-specific parameters $\theta_i'$ by performing a gradient descent step on the initial weights $\theta$ with learning rate $\alpha$. Second, in the *meta-optimization* step (line 8), we evaluate the performance of these adapted parameters

$\theta_i'$. The global initialization $\theta$ is then updated via stochastic gradient descent with learning rate $\beta$, explicitly optimizing the policy's ability to undergo fast adaptation to new tasks.

# E. Detailed Experimental Setup

## E.1. Setup for 3D-BPP

*Table 4.* Hyperparameters of our experiments.

| Hyperparameter | Value | Hyperparameter | Value |
|---|---|---|---|
| Initialization epoch | 250 | Finetune epoch | 50 |
| Number of batches per epoch | 200 | Batch size | 64 |
| Episode length | 70 | Number of GAT Layers | 1 |
| Embedding size | 64 | Hidden size | 128 |
| Number of leaf nodes for discrete environment | 50 | Number of leaf nodes for discrete environment | 100 |
| Number of random seeds | 3 | Action Heuristics for PCT | EMS |
| k | 3 | | |

**Hyperparameters for ASAP.** Table 4 gives the hyperparemters we used during our experiments. To guarantee a fair comparison, all the baseline methods, which also applies GAT architectures also apply the network hyperparameters. All the experiments are performed on the same machine, equipped with a single Intel Core i7-12700 CPU and a single RTX 4090 GPU. Code implementation are included in the supplementary material.

**Hyperparameter for baselines.** To guarantee a fair comparison, We strive to provide environments for each baseline to achieve their best performance. To replicate the optimal results of each baseline, we utilized their suggested default hyperparameters and also tested with 3 different random seeds. For baselines that may have different objectives, such as AR2L, we set their hyperparameters to those that yield the best performance among their reported results. For example, we set $\alpha = 1$ for AR2L to ensure optimal performance for comparison.

**Hyperparameter for environment.** Concerning the environment setup, since both the baselines and our proposed ASAP framework are compatible with the environment developed by PCT, we adhered to the recommended hyperparameters from PCT. For the action-generation heuristics, ASAP demonstrates robust performance when altering the action-generation heuristics. For different types of heuristics, we applied the suggested Empty Maximal Space (EMS) heuristic from PCT. Our method is also compatible with other heuristics like EV, EP, and CP, but no significant improvements compared to EMS are observed. Furthermore, increasing the number of generated actions beyond a certain threshold - 50 for discrete environments and 100 for continuous environments - does not yield meaningful performance gains but instead raised computational costs. Consequently, in our experiments, we use the setting shown in Table 4 for the training and evaluation environments.

**DRL Method Selection.** We also performed experiments with PPO, which is used in AR2L and GOPT, but finally we opted for ACKTR due to its high sample efficiency considering the simulation cost of the BPP environment. Note that ASAP is compatible with all traditional DRL algorithms.

**Dataset.** We follow the setting of PCT but make some small modifications. The item set for each subset in the discrete environment is as follows: Default ($l, w, h \in \{2, 4, 6, 8, 10\}$), ID-Large ($l, w, h \in \{6, 8, 10\}$), ID-Medium ($l, w, h \in \{4, 6, 8\}$), ID-Small ($l, w, h \in \{2, 4, 6\}$). Meanwhile we have out-of-distribution datasets as OOD ($l, w, h \in \{1, 2, 3, 4, 5, 6, 7, 8, 9, 10, 11\}$), OOD-Large ($l, w, h \in \{6, 7, 8, 9, 10, 11\}$) and OOD-Small ($l, w, h \in \{1, 2, 3, 4, 5, 6\}$). Note that the original PCT evaluates on a uniform distribution and treats rotated dimensions (e.g., [2,4,10] and [10,4,2]) as distinct shapes. We removed these redundant shapes and sampled a distinct, non-uniform distribution for every batch of instances. For each dataset, we first randomly sample 100 different item distributions. Each item distribution is used to sample a batch of data, where a batch contains 64 item sequences. For the dataset in the continuous environment, we follow the same procedure as generating dataset in the discrete environment. One additional step for continuous environment is that we will append a 3D noise $l^3 \in [-0.5, 0.5]^3$ to augment the size of the items.

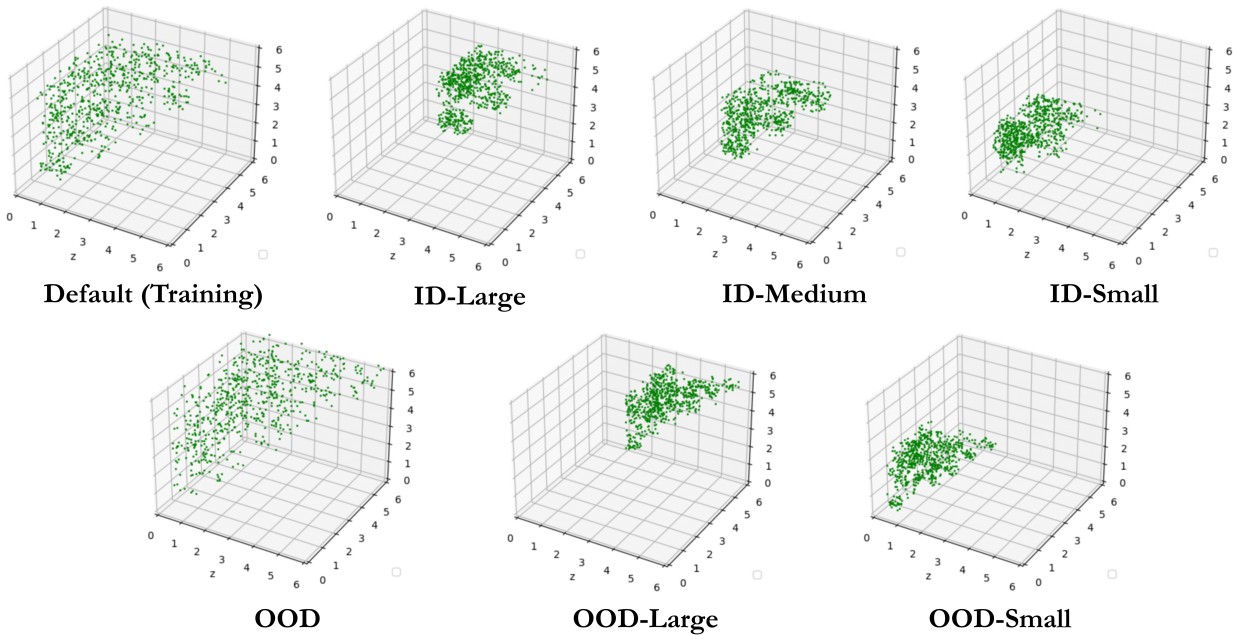

*Figure 8.* Instance distribution for ID and OOD datasets.

### E.2. Setup for TSP/CVRP

To evaluate generalizability across diverse spatial structures, we generate instances following four distinct node distributions, as visualized in Figure 9. All generated instances are normalized to the unit square $[0, 1]^2$. We refer to Bossek et al. (2019a) for further technical specifications.

**Node Distributions.** The specific generation protocols are as follows:

- **Uniform:** Nodes are sampled uniformly at random from the unit square $[0, 1]^2$.

- **Clustered:** We first generate $N$ cluster centers uniformly within a region $[0, L]$. Each node is then assigned to a center and displaced by Gaussian noise with mean $\mu = 1$ and standard deviation $\sigma = 0$. Our datasets comprise a balanced mixture of instances with configurations ($N = 3, L = 10$) and ($N = 7, L = 50$). For CVRP, the depot is placed uniformly alongside the cluster centers.

- **Explosion:** Nodes are initially generated via a uniform distribution. Subsequently, a circular "explosion" region is defined by selecting a center uniformly and a radius $r$ uniformly from $[r_{\min}, r_{\max}]$. All nodes falling within this disc are projected outward according to an exponential distribution with rate $\lambda$. We employ parameters $r_{\min} = 0.1$, $r_{\max} = 0.5$, and $\lambda = 10$.

- **Implosion:** Similar to the explosion distribution, nodes are first placed uniformly, and a disc is defined with radius $r \in [r_{\min}, r_{\max}]$. However, nodes within this disc are projected inward toward the center. The displacement is governed by a multiplier $\lambda \in [1, +\infty)$, drawn from a truncated normal distribution ($\mu = 1, \sigma = 0$). We set $r_{\min} = 0.1$ and $r_{\max} = 0.5$.

*Table 5.* Hyperparameters of TSP/CVRP.

| Hyperparameter | Value | Hyperparameter | Value |
|---|---|---|---|
| Initialization epoch | 400 | Finetune epoch | 50 |
| Number of batches per epoch | 300 | Batch size | 64 |
| Number of Attention Layers for Encoder | 2 | Number of Attention Layers for Decoder | 3 |
| Embedding size | 128 | Feed Forward size | 512 |
| Number of random seeds | 3 | k | 8 |

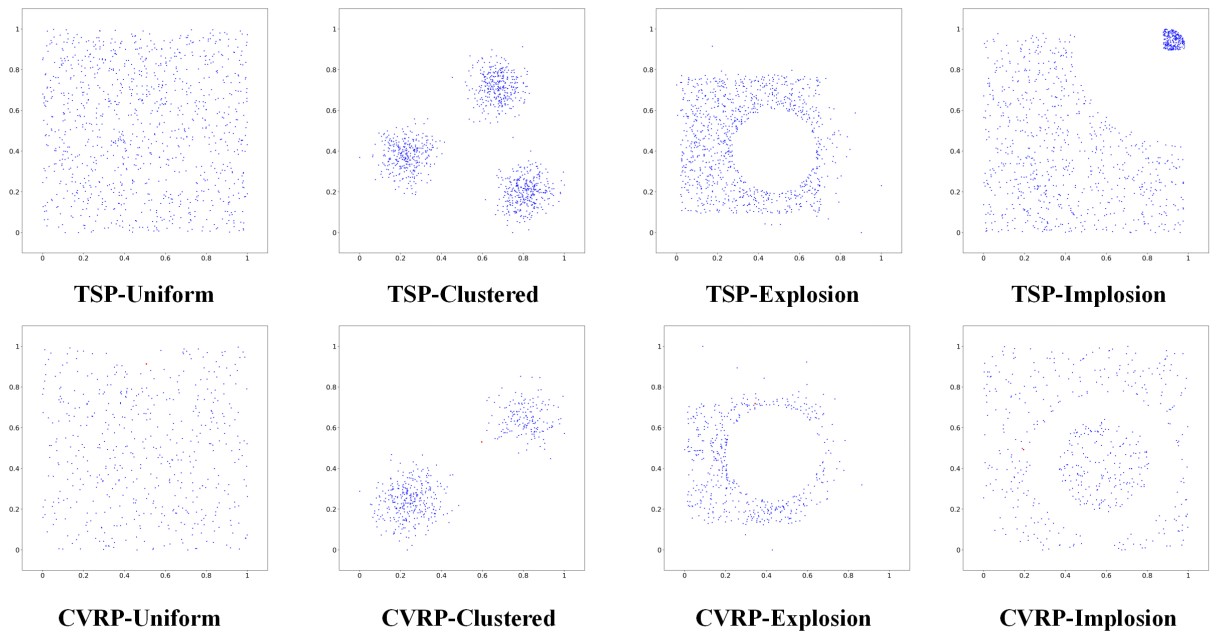

*Figure 9.* Representative problem instances from the evaluation dataset. Each sub-figure illustrates a single instance generated according to the specific distribution protocols defined for TSP1000 and CVRP500. The distributions exhibit varying degrees of spatial structure, ranging from uniform placement to extreme clustering and vacuity.

**CVRP Constraints.**  For CVRP instances following Uniform, Explosion, and Implosion distributions, the depot location is sampled uniformly from the unit square. For Clustered instances, the depot is generated as an additional cluster center. We set the vehicle capacity to $Q = 50$, and the demand for each node is an integer sampled uniformly from the set $\{1, \ldots, 10\}$.

**Ground Truth and Gap Calculation**  As noted in Section 5.1, we employ various solvers to generate (near-)optimal solutions for calculating the optimality gap. The specific configurations are detailed below:

**Traveling Salesman Problem (TSP).**  For small-scale instances (TSP-100), we utilize Gurobi (Gurobi Optimization, LLC, 2023), an exact solver, ensuring that the baselines are truly optimal. For larger scales (TSP-1000 and TSP-10000), exact solving is computationally prohibitive. Instead, we employ LKH3 (Helsgaun, 2009; 2017), a state-of-the-art heuristic. To balance accuracy and computational cost, TSP-1000 instances are solved using 20,000 iterations over 10 runs, while TSP-10000 instances are solved with 20,000 iterations over a single run.

**Capacitated Vehicle Routing Problem (CVRP).**  We utilize the Hybrid Genetic Search (HGS) algorithm (Vidal, 2022) for all CVRP scales. For CVRP-50 and CVRP-500, HGS is run with its default parameter of 20,000 iterations. For the large-scale CVRP-5000 dataset, we impose a strict time limit of 4 hours per instance. We note that this time constraint may result in fewer than 20,000 iterations, leading to slightly sub-optimal ground truth labels. This sub-optimality in the baseline likely contributes to the observation that our method achieves better average gaps on CVRP-5000 compared to CVRP-500. However, as the relative ranking among neural constructive methods is robust to the absolute quality of the ground truth, this approximation is acceptable for comparative evaluation.

**Model Architecture.**  For the INViT baseline, we employ a single-view encoder with 2 attention layers and a decoder with 3 attention layers. For the POMO baseline, we adopt a configuration with 2 encoder layers and 3 decoder layers. Across all models, we set the embedding dimension $d_{model} = 128$ and the feed-forward dimension $d_{ff} = 512$. Each multi-head attention layer consists of 8 heads. Code implementation are included in the supplementary material.

**Training and Environment.**  We use a default batch size of 64 and an augmentation size of 8. The learning rate is initialized to $10^{-4}$. All experiments are conducted using PyTorch 1.12 and Python 3.9. We observed no compatibility issues with these versions for the baseline methods. To mitigate memory fragmentation, the PyTorch GPU split size is capped at 512MB. In rare cases where baselines encounter CUDA out-of-memory (OOM) errors, we dynamically reduce the POMO size (the number of parallel solutions generated) to fit within the available hardware budget.

# F. Additional Experimental Results

*Table 6.* Discrete and Continuous Performance Comparison with and without Online Adaptation on In-distribution Datasets. Approx Optim represents the performance of MCTS.

| | Measurement | Discrete | | | | | | Continuous | | | | | |
| | | w/o Adaptation | | w/ Adaptation | | Improvement | Inference | w/o Adaptation | | w/ Adaptation | | Improvement | Inference |
| | | Uti(%) | Num | Uti(%) | Num | ΔUti(%) | time(m) | Uti(%) | Num | Uti(%) | Num | ΔUti(%) | time(m) |
|---|---|---|---|---|---|---|---|---|---|---|---|---|---|
| **Default** | Approx Optim | 85.9 | 33.7 | / | / | / | / | 66.0 | 24.1 | / | / | / | / |
| | PCT(ICLR-22) | 82.0 | 31.5 | 82.2 | 31.6 | +0.2 | 4.0 | 62.6 | 21.5 | 62.7 | 21.4 | +0.1 | 13.9 |
| | AR2L(Neurips-23) | 80.4 | 29.7 | 80.5 | 29.9 | +0.1 | 4.1 | 61.9 | 20.9 | 62.0 | 20.8 | +0.1 | 13.9 |
| | GOPT(RAL-24) | 80.9 | 30.4 | 81.1 | 30.6 | +0.2 | 4.0 | / | / | / | / | / | / |
| | PCT+ASAP w/o MAML | 83.9 | 32.2 | 84.2 | 32.4 | **+0.3** | 4.2 | 63.7 | 22.2 | 63.9 | 22.3 | **+0.2** | 14.1 |
| | PCT+MAML | 83.6 | 32.0 | 83.8 | 32.1 | +0.2 | 4.1 | 63.5 | 22.1 | 63.5 | 22.1 | +0.0 | 13.9 |
| | **PCT+ASAP w/ MAML**(proposed) | **84.5** | 32.5 | **84.8** | 32.7 | **+0.3** | 4.2 | **64.7** | 22.8 | **64.9** | 22.9 | **+0.2** | 14.1 |
| **ID-Medium** | Approx Optim | 81.5 | 29.3 | / | / | / | / | 67.3 | 25.2 | / | / | / | / |
| | PCT(ICLR-22) | 76.2 | 27.6 | 76.4 | 27.5 | +0.2 | 3.3 | 62.5 | 23.2 | 62.8 | 23.2 | +0.3 | 13.8 |
| | AR2L(Neurips-23) | 72.1 | 26.8 | 72.3 | 27.0 | +0.2 | 3.3 | 61.3 | 23.0 | 61.1 | 23.0 | -0.2 | 13.8 |
| | GOPT(RAL-24) | 75.3 | 27.4 | 75.3 | 27.4 | +0.0 | 3.3 | / | / | / | / | / | / |
| | PCT+ASAP w/o MAML | 78.6 | 28.4 | 79.3 | 28.6 | +0.7 | 3.4 | 63.6 | 23.7 | 64.9 | 23.9 | +0.7 | 13.9 |
| | PCT+MAML | 77.4 | 28.0 | 77.7 | 28.1 | +0.3 | 3.3 | 64.3 | 23.8 | 64.4 | 23.8 | +0.1 | 13.8 |
| | **ASAP**(proposed) | **79.1** | 28.5 | **79.9** | 28.8 | **+0.8** | 3.4 | **65.5** | 24.1 | **66.3** | 24.3 | **+0.8** | 13.9 |
| **OOD** | Approx Optim | 68.7 | 21.6 | / | / | / | / | 61.6 | 18.2 | / | / | / | / |
| | PCT(ICLR-22) | 62.6 | 19.0 | 62.5 | 19.1 | -0.1 | 4.5 | 56.2 | 16.4 | 56.3 | 16.4 | +0.1 | 14.2 |
| | AR2L(Neurips-23) | 62.9 | 19.2 | 63.1 | 19.3 | +0.2 | 4.5 | 56.1 | 16.5 | 56.3 | 16.4 | +0.2 | 14.2 |
| | GOPT(RAL-24) | 62.6 | 19.0 | 62.9 | 19.2 | +0.3 | 4.5 | / | / | / | / | / | / |
| | PCT+ASAP w/o MAML | 63.1 | 19.4 | 65.0 | 20.2 | **+1.9** | 4.6 | 58.5 | 17.1 | 60.3 | 17.6 | **+1.8** | 14.3 |
| | PCT+MAML | 63.6 | 19.5 | 63.9 | 19.7 | +0.3 | 4.5 | 58.1 | 16.9 | 58.4 | 17.0 | +0.3 | 14.2 |
| | **ASAP**(proposed) | **64.1** | 19.8 | **65.6** | 20.3 | +1.5 | 4.6 | **59.5** | 17.4 | **61.0** | 17.8 | +1.5 | 14.3 |

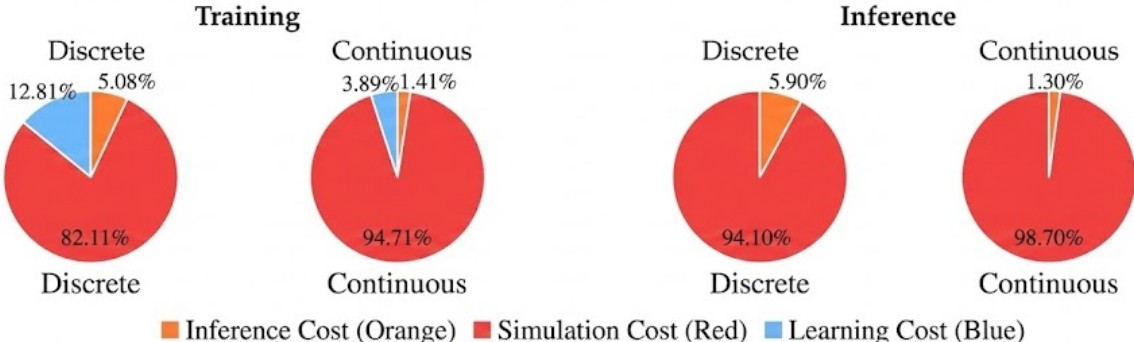

*Figure 10.* Time cost proportion of ASAP for each processes in training and inference.

## F.1. Additional Results for 3D-BPP

In this section, we present a comprehensive analysis of the proposed method's performance across In-Distribution (ID) and Out-of-Distribution (OOD) datasets. We compare our approach against state-of-the-art baselines, including PCT (ICLR-22), AR2L (Neurips-23), and GOPT (RAL-24). Furthermore, we provide a sensitivity analysis regarding the proposal action set size and a computational cost analysis.

*Table 7.* Sensitivity Analysis on In-distribution Datasets. $k$ represents the size of the proposal action set.

| | | Discrete | | | | | | Continuous | | | | |
|---|---|---|---|---|---|---|---|---|---|---|---|---|
| | Measurement | w/o Adaptation Uti(%) | Num | w/ Adaptation Uti(%) | Num | Improvement ΔUti(%) | Inference time(m) | w/o Adaptation Uti(%) | Num | w/ Adaptation Uti(%) | Num | Improvement ΔUti(%) | Inference time(m) |

**Default**

| Measurement | Uti(%) | Num | Uti(%) | Num | ΔUti(%) | time(m) | Uti(%) | Num | Uti(%) | Num | ΔUti(%) | time(m) |
|---|---|---|---|---|---|---|---|---|---|---|---|---|
| PCT+ASAP w/ MAML($k = 1$) | 83.6 | 32.0 | 83.8 | 32.1 | +0.2 | 4.1 | 63.5 | 22.1 | 63.5 | 22.1 | +0.0 | 13.9 |
| PCT+ASAP w/ MAML($k = 3$) | **84.5** | 32.5 | **84.8** | 32.7 | **+0.3** | 4.2 | 64.0 | 22.4 | 64.2 | 22.5 | +0.2 | 14.1 |
| PCT+ASAP w/ MAML($k = 5$) | 84.3 | 32.4 | 84.5 | 32.7 | +0.2 | 4.2 | 64.3 | 22.6 | 64.5 | 22.7 | +0.2 | 14.1 |
| PCT+ASAP w/ MAML($k = 8$) | 84.2 | 32.3 | 84.5 | 32.7 | +0.3 | 4.2 | 64.6 | 22.7 | **64.9** | 22.9 | **+0.3** | 14.1 |
| PCT+ASAP w/ MAML($k = 10$) | 84.2 | 32.3 | 84.4 | 32.7 | +0.2 | 4.2 | **64.7** | 22.8 | **64.9** | 22.9 | +0.2 | 14.1 |

**Large**

| Measurement | Uti(%) | Num | Uti(%) | Num | ΔUti(%) | time(m) | Uti(%) | Num | Uti(%) | Num | ΔUti(%) | time(m) |
|---|---|---|---|---|---|---|---|---|---|---|---|---|
| PCT+ASAP w/ MAML($k = 1$) | 71.7 | 11.5 | 72.0 | 11.6 | +0.3 | 1.1 | 62.3 | 9.9 | 62.3 | 9.9 | +0.0 | 5.8 |
| PCT+ASAP w/ MAML($k = 3$) | **73.5** | 12.1 | **74.2** | 12.3 | **+0.7** | 1.1 | 62.8 | 10.2 | 63.7 | 10.3 | +0.9 | 5.9 |
| PCT+ASAP w/ MAML($k = 5$) | 73.4 | 12.1 | **74.2** | 12.3 | **+0.8** | 1.1 | **63.2** | 10.2 | **64.1** | 10.3 | **+0.9** | 5.9 |
| PCT+ASAP w/ MAML($k = 8$) | 73.4 | 12.1 | 74.0 | 12.2 | +0.6 | 1.1 | 63.0 | 10.2 | 63.8 | 10.3 | +0.8 | 5.9 |
| PCT+ASAP w/ MAML($k = 10$) | **73.5** | 12.1 | 74.1 | 12.2 | +0.6 | 1.1 | 63.1 | 10.2 | 63.9 | 10.3 | +0.8 | 5.9 |

**Medium**

| Measurement | Uti(%) | Num | Uti(%) | Num | ΔUti(%) | time(m) | Uti(%) | Num | Uti(%) | Num | ΔUti(%) | time(m) |
|---|---|---|---|---|---|---|---|---|---|---|---|---|
| PCT+ASAP w/ MAML($k = 1$) | 77.4 | 28.0 | 77.7 | 28.1 | +0.3 | 3.3 | 64.3 | 23.8 | 64.4 | 23.8 | +0.1 | 13.8 |
| PCT+ASAP w/ MAML($k = 3$) | **79.1** | 28.5 | **79.9** | 28.8 | **+0.8** | 3.4 | 64.8 | 24.0 | 65.7 | 24.2 | **+0.9** | 13.9 |
| PCT+ASAP w/ MAML($k = 5$) | 78.7 | 28.3 | 79.6 | 28.7 | **+0.9** | 3.4 | 65.0 | 24.1 | 65.8 | 24.2 | +0.8 | 13.9 |
| PCT+ASAP w/ MAML($k = 8$) | 79.0 | 28.5 | **79.9** | 28.8 | **+0.9** | 3.4 | 65.4 | 24.1 | **66.2** | 24.3 | **+0.8** | 13.9 |
| PCT+ASAP w/ MAML($k = 10$) | 78.5 | 28.3 | 79.3 | 28.6 | +0.8 | 3.4 | **65.5** | 24.1 | **66.3** | 24.3 | +0.8 | 13.9 |

**Small**

| Measurement | Uti(%) | Num | Uti(%) | Num | ΔUti(%) | time(m) | Uti(%) | Num | Uti(%) | Num | ΔUti(%) | time(m) |
|---|---|---|---|---|---|---|---|---|---|---|---|---|
| PCT+ASAP w/ MAML($k = 1$) | 85.8 | 98.2 | 86.0 | 98.3 | +0.2 | 12.0 | 66.8 | 80.8 | 66.9 | 80.8 | +0.1 | 42.0 |
| PCT+ASAP w/ MAML($k = 3$) | **86.5** | 99.0 | **87.4** | 101.0 | +0.9 | 12.2 | 67.0 | 80.7 | 67.7 | 81.5 | +0.7 | 42.3 |
| PCT+ASAP w/ MAML($k = 5$) | **86.5** | 99.0 | 87.2 | 100.8 | +0.7 | 12.2 | 67.3 | 81.3 | 67.9 | 81.7 | +0.6 | 42.3 |
| PCT+ASAP w/ MAML($k = 8$) | 86.1 | 98.9 | 87.0 | 101.0 | **+0.9** | 12.2 | 67.3 | 81.3 | 68.1 | 81.8 | **+0.8** | 42.3 |
| PCT+ASAP w/ MAML($k = 10$) | 86.1 | 98.7 | 86.9 | 100.5 | +0.8 | 12.2 | **67.6** | 81.4 | **68.3** | 81.9 | +0.7 | 42.3 |

### F.1.1. PERFORMANCE COMPARISON

Table 6 summarizes the performance of our proposed method (**PCT+ASAP w/ MAML**) against baselines in both Discrete and Continuous action spaces.

**In-Distribution (ID) Performance.** On standard ID datasets (Default and Medium), our method consistently outperforms existing baselines.

- **Discrete Control:** In the default setting, our method achieves a utilization of **84.8%** with adaptation, surpassing the strongest baseline (PCT) which achieves 82.2%. Similarly, on the ID-Medium dataset, our method demonstrates a significant lead, reaching **79.9%** utilization compared to 76.4% for PCT.

- **Continuous Control:** The trend continues in continuous spaces. Our method achieves **64.9%** utilization in the default setting, significantly outperforming AR2L (62.0%) and PCT (62.7%).

- **Effectiveness of Adaptation:** The improvement (ΔUti) contributed by the online adaptation module is consistently positive. For instance, in the ID-Medium discrete task, adaptation yields a **+0.8%** boost for our method, whereas baselines like GOPT show negligible improvement ($+0.0\%$).

**Out-of-Distribution (OOD) Robustness.** The advantages of our proposed meta-learning framework are most pronounced in OOD scenarios, where the test distribution shifts from the training distribution.

- As shown in the OOD section of Table 6, our method achieves an improvement of **+1.5%** via adaptation in the discrete setting, boosting utilization from 64.1% to **65.6%**. This is a substantial gain compared to the baselines (e.g., PCT shows a slight regression of -0.1%, and AR2L improves by only +0.2%).

- This empirical evidence validates that embedding MAML-based adaptation significantly enhances generalization capabilities in unseen environments compared to standard fine-tuning or heuristic adaptation methods.

*Table 8.* Sensitivity Analysis on Out-of-distribution Datasets. $k$ represents the size of the proposal action set.

| | | Discrete | | | | | | Continuous | | | | | |
|---|---|---|---|---|---|---|---|---|---|---|---|---|---|
| | Measurement | w/o Adaptation | | w/ Adaptation | | Improvement | Inference | w/o Adaptation | | w/ Adaptation | | Improvement | Inference |
| | | Uti(%) | Num | Uti(%) | Num | $\Delta$Uti(%) | time(m) | Uti(%) | Num | Uti(%) | Num | $\Delta$Uti(%) | time(m) |
| **OOD** | PCT+ASAP w/ MAML($k=1$) | 63.6 | 19.5 | 63.9 | 19.7 | +0.3 | 4.5 | 58.1 | 16.9 | 58.4 | 17.0 | +0.3 | 14.2 |
| | PCT+ASAP w/ MAML($k=3$) | 64.1 | 19.8 | **65.6** | 20.3 | **+1.5** | 4.6 | 58.8 | 17.2 | 60.2 | 17.4 | +1.4 | 14.3 |
| | PCT+ASAP w/ MAML($k=5$) | 64.1 | 19.8 | 65.5 | 20.3 | +1.4 | 4.6 | 59.0 | 17.3 | 60.3 | 17.4 | +1.3 | 14.3 |
| | PCT+ASAP w/ MAML($k=8$) | 63.8 | 19.6 | 65.0 | 20.1 | +1.2 | 4.6 | **59.7** | 17.5 | **61.0** | 17.8 | +1.3 | 14.3 |
| | PCT+ASAP w/ MAML($k=10$) | **64.3** | 19.8 | 65.2 | 20.3 | +0.9 | 4.6 | 59.5 | 17.4 | **61.0** | 17.8 | **+1.5** | 14.3 |
| | Measurement | w/o Adaptation | | w/ Adaptation | | Improvement | Inference | w/o Adaptation | | w/ Adaptation | | Improvement | Inference |
| | | Uti(%) | Num | Uti(%) | Num | $\Delta$Uti(%) | time(m) | Uti(%) | Num | Uti(%) | Num | $\Delta$Uti(%) | time(m) |
| **OOD-Large** | PCT+ASAP w/ MAML($k=1$) | 60.8 | 6.7 | 61.0 | 6.7 | +0.2 | 1.0 | 54.5 | 6.1 | 54.8 | 6.2 | +0.3 | 5.0 |
| | PCT+ASAP w/ MAML($k=3$) | **61.5** | 6.7 | **62.4** | 6.9 | **+0.9** | 1.0 | 55.3 | 6.2 | 56.3 | 6.3 | +1.0 | 5.1 |
| | PCT+ASAP w/ MAML($k=5$) | **61.5** | 6.7 | **62.4** | 6.9 | **+0.9** | 1.0 | 55.5 | 6.2 | 56.7 | 6.4 | +1.2 | 5.1 |
| | PCT+ASAP w/ MAML($k=8$) | 61.0 | 6.7 | 61.8 | 6.8 | +0.8 | 1.0 | **55.7** | 6.2 | **57.0** | 6.4 | **+1.3** | 5.1 |
| | PCT+ASAP w/ MAML($k=10$) | 61.2 | 6.7 | 62.0 | 6.8 | +0.8 | 1.0 | **55.7** | 6.2 | **57.0** | 6.4 | **+1.3** | 5.1 |
| | Measurement | w/o Adaptation | | w/ Adaptation | | Improvement | Inference | w/o Adaptation | | w/ Adaptation | | Improvement | Inference |
| | | Uti(%) | Num | Uti(%) | Num | $\Delta$Uti(%) | time(m) | Uti(%) | Num | Uti(%) | Num | $\Delta$Uti(%) | time(m) |
| **OOD-Small** | PCT+ASAP w/ MAML($k=1$) | 81.2 | 145.0 | 82.1 | 146.2 | +0.9 | 16.8 | 69.6 | 119.4 | 70.1 | 120.8 | +0.5 | 47.3 |
| | PCT+ASAP w/ MAML($k=3$) | 82.3 | 146.5 | **84.5** | 148.8 | +2.2 | 17.1 | 69.8 | 120.0 | 71.9 | 122.8 | +2.1 | 47.6 |
| | PCT+ASAP w/ MAML($k=5$) | 82.1 | 146.1 | 84.4 | 148.6 | +2.3 | 17.1 | 70.0 | 120.4 | 72.3 | 123.8 | **+2.3** | 47.6 |
| | PCT+ASAP w/ MAML($k=8$) | 82.0 | 146.0 | 84.0 | 148.0 | +2.0 | 17.1 | 70.4 | 121.1 | 72.3 | 123.8 | +1.9 | 47.6 |
| | PCT+ASAP w/ MAML($k=10$) | **82.4** | 146.7 | 83.8 | 147.8 | +1.4 | 17.1 | **70.5** | 121.3 | **72.6** | 124.2 | +2.1 | 47.6 |

*Table 9.* Training time cost and average adaptation time cost

| Method | Discrete | | Continuous | |
|---|---|---|---|---|
| | Training | Adaptation | Training | Adaptation |
| PCT(ICLR-22) | 1d21h | 6.5m | 6d6h | 22.3m |
| AR2L(Neurips-23) | 1d22h | 6.6m | 6d8h | 22.5m |
| GOPT(RAL-24) | 1d21h | 6.6m | 6d7h | 22.4m |
| **PCT+ASAP w/ MAML**(proposed) | Pre-training: 1d14h post-training: 8h    total: 1d22h | 6.5m | pre-training: 5d6h post-training: 1d2h    total: 6d8h | 22.2m |

### F.1.2. SENSITIVITY ANALYSIS

Tables 7 and 8 present an ablation study on the hyperparameter $k$ (size of the proposal action set) across different dataset scales (Large, Medium, Small).

- **Impact of $k$:** There is a clear positive correlation between $k$ and performance. Increasing $k$ from 1 to 3 yields the most significant marginal gain. For example, in the OOD-Small Discrete setting (Table 8), increasing $k$ from 1 to 3 improves the adaptation gain from +0.9% to +2.2%.

- **Saturation:** Performance gains tend to saturate around $k=8$ or $k=10$. In the ID-Default setting (Table 7), the difference between $k=3$ (84.8%) and $k=10$ (84.4%) is negligible or slightly regressive, suggesting that a small proposal set ($k=3$ to 5) is sufficient to capture diverse, high-quality candidates without incurring excessive computational overhead.

- **Adaptation Stability:** The improvement metric ($\Delta$Uti) remains robust across different scales, consistently showing positive values, which indicates the stability of the proposed adaptation mechanism regardless of problem size.

### F.1.3. COMPUTATIONAL EFFICIENCY

We analyze the computational costs associated with our method in Table 9 and the provided pie charts.

- **Training Overhead:** Although our method introduces a meta-learning phase, the total training time remains competitive. As shown in Table 9, the total discrete training time (1d 22h) matches that of AR2L and is only marginally higher than PCT (1d 21h). This is because the additional post-training phase (8h) is offset by efficient pre-training.

- **Inference and Adaptation:** The online adaptation time (6.5m for Discrete) is identical to the baselines, ensuring that our performance gains do not come at the cost of deployment latency.

- **Cost Breakdown:** The pie charts in Figure 1 (Context) reveal that *Simulation Cost* dominates the computational budget (approx. 82%-98% across settings), while *Learning Cost* is a minor fraction (approx. 1.4%-12.8%). This justifies the use of slightly more complex learning algorithms (like MAML), as they introduce negligible overhead relative to the simulation bottleneck while significantly improving policy quality.

## F.2. Additional Results for TSP/CVRP

*Table 10.* Comparison of Optimality Gaps (Pre- and Post-Adaptation) for POMO and INViT-1V Policies w/ and w/o ASAP across TSPLIB and CVRPLIB.

| Policy | 1 ~ 100 | | | 100 ~ 1000 | | | > 1000 | | |
|---|---|---|---|---|---|---|---|---|---|
| | Before | After | Imp. | Before | After | Imp. | Before | After | Imp. |
| POMO | 1.92% | 1.85% | 0.07% | 13.49% | 12.75% | 0.74% | 60.05% | 71.22% | -11.17% |
| POMO+ASAP w/o MAML | 1.28% | 1.03% | **0.25%** | 10.37% | 8.78% | 1.59% | 51.25% | 44.53% | 6.72% |
| POMO+MAML | 1.47% | 1.32% | 0.15% | 10.11% | 9.12% | 0.99% | 53.21% | 49.28% | 3.93% |
| POMO+ASAP w/ MAML | **1.11%** | **0.99%** | 0.12% | **9.43%** | **7.72%** | **1.71%** | **47.45%** | **40.13%** | **7.32%** |
| INViT-1V | 2.65% | 2.41% | 0.24% | 6.22% | 5.73% | 0.49% | 10.33% | 9.77% | 0.56% |
| INViT-1V+ASAP w/o MAML | 1.36% | 1.07% | 0.29% | 3.95% | 3.02% | **0.93%** | 8.36% | 7.41% | **0.95%** |
| INViT-1V+MAML | 1.77% | 1.32% | **0.45%** | 4.67% | 4.07% | 0.60% | 9.75% | 9.10% | 0.65% |
| INViT-1V+ASAP w/ MAML | **1.07%** | **0.99%** | 0.08% | **3.42%** | **2.79%** | 0.63% | **7.67%** | **6.98%** | 0.69% |

TSPLIB (left row label rotated)

| Policy | 101 ~ 200 | | | 201 ~ 500 | | | > 500 | | |
|---|---|---|---|---|---|---|---|---|---|
| | Before | After | Imp. | Before | After | Imp. | Before | After | Imp. |
| POMO | 9.43% | 9.02% | 0.41% | 19.76% | 18.97% | 0.79% | 58.82% | 72.33% | -13.51% |
| POMO+ASAP w/o MAML | **8.21%** | 7.13% | 1.08% | 16.22% | 14.07% | **2.15%** | 50.21% | 42.29% | **7.92%** |
| POMO+MAML | 8.73% | 8.16% | 0.57% | 16.95% | 15.78% | 1.17% | 53.75% | 49.23% | 4.52% |
| POMO+ASAP w/ MAML | 8.35% | **7.09%** | 1.26% | **15.01%** | **13.75%** | 1.26% | **46.78%** | **39.36%** | 7.42% |
| INViT-1V | 10.35% | 9.43% | 0.92% | 14.27% | 13.12% | 1.15% | 19.35% | 17.66% | 1.69% |
| INViT-1V+ASAP w/o MAML | 9.18% | 7.92% | **1.26%** | 12.36% | 10.27% | **2.09%** | 14.81% | 12.75% | **2.06%** |
| INViT-1V+MAML | 9.62% | 8.71% | 0.91% | 12.21% | 11.45% | 0.76% | 16.57% | 15.04% | 1.53% |
| INViT-1V+ASAP w/ MAML | **8.19%** | **6.94%** | 1.25% | **11.05%** | **9.79%** | 1.26% | **12.19%** | **10.32%** | 1.87% |

CVRPLIB Set-X (left row label rotated)

### F.2.1. ADDITIONAL RESULTS ON TSPLIB AND CVRPLIB

Table 10 evaluates the pre- and post-adaptation optimality gaps of POMO and INViT-1V on TSPLIB and CVRPLIB benchmarks across varying instance sizes. The baseline POMO policy exhibits severe generalization challenges on larger instances, notably deteriorating after standard adaptation on TSPLIB scales $> 1000$ (a $-11.17\%$ improvement) and CVRPLIB scales $> 500$ ($-13.51\%$). Integrating ASAP (ASAP w/ MAML) completely mitigates this degradation, yielding the lowest initial optimality gaps and ensuring consistent, positive post-adaptation gains across all evaluated scales. Furthermore, applying ASAP to the INViT-1V backbone establishes the most robust overall performance, restricting the post-adaptation gap to just 6.98% on TSPLIB ($> 1000$) and 10.32% on CVRPLIB ($> 500$). These results demonstrate that the proposed framework successfully stabilizes and enhances online adaptation, particularly for challenging, large-scale out-of-distribution routing problems.

### F.2.2. SENSITIVITY ANALYSIS

**Sensitivity to Proposal Action Set Size ($k$).** Table 11 investigates the impact of the proposal action set size, denoted as $k \in \{8, 10, 15\}$, on the adaptation performance of the INViT-1V+ASAP w/o MAML policy. The results demonstrate that the method exhibits a high degree of robustness to variations in $k$ across all tested distributions (Clustered, Implosion, and Explosion). For instance, on the *Implosion* dataset for TSP-1000, the post-adaptation optimality gaps for INViT-1V remain remarkably stable at **6.99%** ($k = 8$), **7.14%** ($k = 10$), and **7.35%** ($k = 15$). Similarly, in the *Clustered* scenario for TSP-100, the variation in optimality gaps is marginal, ranging only from **3.67%** to **4.11%**. This consistency suggests that the ASAP mechanism effectively leverages the available proposal actions to refine the policy, regardless of slight changes in the proposal set size. The comprehensive analysis across both TSP and CVRP tasks confirms that our method is **not**

*Table 11.* Sensitive analysis for INViT-1V+ASAP w/o MAML and POMO+ASAP w/o MAML policies across Clustered, Implosion, and Explosion datasets for TSP and CVRP.

| Policy | FT: TSP-100 / Eval: TSP-100 | | | | FT: TSP-100 / Eval: TSP-1000 | | | | FT: CVRP-50 / Eval: CVRP-50 | | | | FT: CVRP-100 / Eval: CVRP-500 | | | |
|---|---|---|---|---|---|---|---|---|---|---|---|---|---|---|---|---|
| | Before | After | Imp. | Time | Before | After | Imp. | Time | Before | After | Imp. | Time | Before | After | Imp. | Time |
| **Clustered** | | | | | | | | | | | | | | | | |
| (Near-)Optimality | 0.00% | / | / | 3.4m | 0.00% | / | / | 16.9h | 0.00% | / | / | 34.2m | 0.00% | / | / | 2.3d |
| POMO+ASAP w/ MAML (k=8) | **4.15%** | 3.82% | 0.33% | 0.3m | 46.15% | 38.73% | 7.42% | 2.3m | **6.72%** | 5.33% | 1.39% | 0.5m | 19.73% | 15.35% | **4.38%** | 3.7m |
| POMO+ASAP w/ MAML (k=10) | 4.29% | 3.85% | **0.44%** | 0.3m | 49.33% | **36.63%** | **12.70%** | 2.3m | 7.50% | **4.79%** | **2.71%** | 0.5m | **16.63%** | **14.82%** | 1.81% | 3.7m |
| POMO+ASAP w/ MAML (k=15) | 4.22% | 3.84% | 0.38% | 0.3m | **45.40%** | 39.06% | 6.34% | 2.4m | 6.78% | 4.90% | 1.88% | 0.5m | 17.51% | 14.86% | 2.65% | 3.8m |
| INViT-1V+ASAP w/ MAML (k=8) | 4.45% | 4.11% | 0.34% | 0.5m | 11.13% | 7.94% | 3.19% | 5.3m | 5.43% | 3.78% | **1.65%** | 1.1m | 11.37% | 9.15% | **2.22%** | 10.7m |
| INViT-1V+ASAP w/ MAML (k=10) | 4.69% | 3.84% | **0.85%** | 0.5m | **10.75%** | 7.44% | 3.31% | 5.3m | **4.92%** | 3.97% | 0.95% | 1.2m | 12.39% | 10.51% | 1.88% | 10.8m |
| INViT-1V+ASAP w/ MAML (k=15) | **4.11%** | **3.67%** | 0.44% | 0.5m | 11.83% | **7.20%** | 4.63% | 5.5m | 5.23% | **4.62%** | 0.61% | 1.3m | **11.21%** | **9.46%** | 1.75% | 11.0m |
| **Implosion** | | | | | | | | | | | | | | | | |
| (Near-)Optimality | 0.00% | / | / | 2.8m | 0.00% | / | / | 17.5h | 0.00% | / | / | 30.0m | 0.00% | / | / | 2.0d |
| POMO+ASAP w/ MAML (k=8) | **2.21%** | 2.05% | 0.16% | 0.3m | 56.73% | 52.11% | 4.62% | 2.3m | 6.35% | 4.61% | **1.74%** | 0.5m | 19.35% | 14.79% | **4.56%** | 3.7m |
| POMO+ASAP w/ MAML (k=10) | 2.45% | **1.58%** | **0.87%** | 0.3m | 54.73% | 50.21% | 4.53% | 2.3m | **5.56%** | 4.17% | 1.39% | 0.5m | 18.23% | 14.15% | 4.08% | 3.7m |
| POMO+ASAP w/ MAML (k=15) | 2.29% | 1.96% | 0.33% | 0.3m | 56.24% | 51.12% | **5.13%** | 2.4m | **5.56%** | 3.99% | 1.57% | 0.5m | **15.65%** | **13.91%** | 1.74% | 3.8m |
| INViT-1V+ASAP w/ MAML (k=8) | 3.53% | **2.69%** | 0.84% | 0.5m | 9.67% | **6.99%** | **2.68%** | 5.3m | 4.92% | 3.53% | **1.39%** | 1.1m | 12.23% | 9.49% | 2.74% | 10.7m |
| INViT-1V+ASAP w/ MAML (k=10) | 3.86% | 2.84% | **1.03%** | 0.5m | **9.52%** | 7.14% | 2.38% | 5.3m | 4.63% | 3.75% | 0.88% | 1.2m | 13.13% | 10.97% | 2.16% | 10.8m |
| INViT-1V+ASAP w/ MAML (k=15) | **3.42%** | 2.72% | 0.69% | 0.5m | 9.62% | 7.35% | 2.27% | 5.4m | **4.54%** | **3.20%** | 1.33% | 1.2m | **11.83%** | **8.64%** | **3.19%** | 11.0m |
| **Explosion** | | | | | | | | | | | | | | | | |
| (Near-)Optimality | 0.00% | / | / | 2.9m | 0.00% | / | / | 17.5h | 0.00% | / | / | 30.0m | 0.00% | / | / | 2.8d |
| POMO+ASAP w/ MAML(k=8) | **2.65%** | 2.47% | 0.18% | 0.3m | 50.35% | 44.26% | 6.09% | 2.3m | 6.25% | 4.43% | 1.82% | 0.5m | 19.78% | **14.33%** | 5.45% | 3.7m |
| POMO+ASAP w/ MAML(k=10) | 2.92% | 2.61% | 0.31% | 0.3m | 52.41% | 48.42% | 3.99% | 2.3m | **5.56%** | 5.16% | 0.40% | 0.5m | 21.03% | 14.58% | **6.45%** | 3.7m |
| POMO+ASAP w/ MAML(k=15) | 2.91% | **2.21%** | **0.70%** | 0.3m | **49.33%** | **41.52%** | **7.82%** | 2.4m | 6.46% | **3.99%** | **2.47%** | 0.5m | **18.06%** | 15.35% | 2.71% | 3.8m |
| INViT-1V+ASAP w/ MAML(k=8) | 3.37% | 2.31% | 1.06% | 0.5m | **10.35%** | 8.27% | 2.08% | 5.3m | 5.29% | 4.15% | 1.14% | 1.1m | 12.19% | 8.95% | 3.24% | 10.8m |
| INViT-1V+ASAP w/ MAML(k=10) | 3.30% | **2.14%** | **1.17%** | 0.5m | 11.83% | 8.52% | **3.31%** | 5.4m | 6.13% | 3.96% | **2.17%** | 1.2m | **11.75%** | **7.46%** | **4.29%** | 10.9m |
| INViT-1V+ASAP w/ MAML(k=15) | **3.01%** | 2.22% | 0.79% | 0.5m | 10.84% | **7.63%** | 3.21% | 5.5m | **4.68%** | **3.74%** | 0.93% | 1.2m | 12.56% | 9.39% | 3.17% | 11.1m |

**sensitive** to the hyperparameter $k$.

*Table 12.* Training time cost and average adaptation time cost for Routing Problem

| Method | TSP-100 | | | CVRP-50 | | |
|---|---|---|---|---|---|---|
| | Training | | Adaptation | Training | | Adaptation |
| POMO | 5h50m | | 20.0m | 6h55m | | 21.2m |
| INViT | 2d2h | | 53.2m | 1d3h | | 40.5m |
| **POMO+ASAP w/ MAML**(proposed) | Pre-training: 3h50m post-training: 2h32m | total: 6h22m | 21.5m | pre-training: 4h11m post-training: 2h56m | total: 7h7m | 22.8m |
| **INViT+ASAP w/ MAML**(proposed) | Pre-training: 1d3h post-training: 15h | total: 2d4h | 56.1m | pre-training: 17h post-training: 12h | total: 1d5h | 41.7m |

### F.2.3. COMPUTATIONAL EFFICIENCY

**Computational Efficiency Analysis.** Table 12 presents the training and adaptation time costs for the TSP-100 and CVRP-50 tasks. A key advantage of our proposed ASAP framework is its computational efficiency, as it introduces negligible time overhead compared to the baseline policies. For instance, when applying our method to the POMO policy on CVRP-50, the total training time increases only marginally from **6h 55m** to **7h 7m**, an addition of just 12 minutes. Similarly, for the more computationally intensive INViT policy on TSP-100, the total training duration rises from **2d 2h** to **2d 4h**, representing a minimal increase relative to the overall training timeline.

**Adaptation Overhead.** The adaptation phase mirrors this trend of efficiency. The time required for adaptation with ASAP remains highly comparable to the baselines, with increases typically measured in mere minutes. For example, adapting the INViT+ASAP w/ MAML policy on TSP-100 takes **56.1m** compared to **53.2m** for the baseline, and on CVRP-50, the difference is only roughly 1 minute (**41.7m** vs **40.5m**). Overall, these results demonstrate that ASAP enhances the model's capabilities without imposing significant computational burdens. The slight additional time cost is a worthwhile trade-off for the performance gains achieved, making the method practical for training and deployment in resource-constrained environments.

