# OpenReview forum: "ASAP: Exploiting the Satisficing Generalization Edge in Neural Combinatorial Optimization"
_ICML.cc/2026/Conference — ICML 2026 regular_

### Official Review · Reviewer_VQqm · 2026-02-18

**Soundness:** 3
**Presentation:** 2
**Significance:** 3
**Originality:** 3
**Overall Recommendation:** 3
**Confidence:** 3

**Summary:**

This paper proposes Adaptive Selection After Proposal (ASAP) method, which divides the problem-solving process from one-stage action selection to two-stage proposal-then-selection. Based on the observation that good proposals are similar across different distributions and the theoretical demonstration that the two-stage process has a higher probability of selecting the optimal action than a single-stage process under specific assumptions, ASAP only updates the selection policy during fine-tuning on a new distribution. On TSP, CVRP and 3D-BPP, ASAP improves the performance of existing solvers.

**Compliance With Llm Reviewing Policy:**

Affirmed.

**Final Justification:**

I find the core idea of ASAP interesting. However, I do not find the reported gains sufficiently convincing given the substantial increase in model size and the "garbage-in, garbage-out" risk introduced by the two-policy design. Overall, I think this work is based on an interesting idea, but it still feels somewhat immature in its current form.

**Key Questions For Authors:**

1. Given an existing solver (e.g., POMO), what is the model architecture of the proposal policy and the selection policy in ASAP? How can they be initialized by a single one-stage model while they have different inputs and outputs? And how many additional parameters are introduced by ASAP?
2. What is the loss function to update the two policies? Are they updated simultaneously or iteratively?
3. Would adapting both policies lead to better performance? Especially under situations where negative improvement occurs. Answers should be supported by empirical evidence.

**Limitations:**

No. A more comprehensive analysis would benefit from additional discussion oflimitations related to (i) the number of extra parameters introduced by the two-policy design and (ii) the extent of distribution shift beyond which the proposal policy must also be adapted (rather than adapting only the selection policy).

**Strengths And Weaknesses:**

Strengths:
* This paper proposes a novel two-stage proposal-then-selection paradigm, with sufficient theoretical and empirical evidence to support the motivation.
* Comprehensive experiments are presented, across different problems, distributions and scales, to verify the effectiveness of the proposed method.
* The background review is clear and well organized, covering the key directions on combinatorial RL generalization and adaptation.

Weaknesses:
* Presentation needs significant improvement. Multiple key components of the proposed method are not clearly described, including
    * the loss function to update the two policies
    * the architecture of the two policies, and how the two policies with different inputs and outputs can be initialized by a single One-Stage model
    * how to embed ASAP to existing solvers (e.g., should both encoder and decoder or only the decoder be duplicated to form two policies), like PCT and POMO, and how many additional model parameters are introduced
    * $p_i(\mathcal{I})$ is defined based on items (of 3D-BPP), how it applied to routing problems?
    * Fig. 3 implies that, during tuning, the two policies are updated interactively (one is frozen when another is updated). But the main text (Section 4.2) implies the two policies are updated simultaneously
    * Fig. 3. Environment exists in A inference flow but not in C adaptation, does this imply that $\mathcal{L}_{adapt}$ is irrelevant to the environment?
* Figures are all blurry
* Counterintuitive experimental observations are not well discussed: Table 2, the improvement of some policies under some situations is negative after adaptation. Does this mean that the distribution difference is too large to transfer useful knowledge? Should the proposal policy still be frozen during adaptation under these situations?
* Fairness of comparison: based on the description in Section 4.2, MAML involves additional problem-solving (trajectories collection) processes, so is it fair to compare w/ MAML  with w/o MAML versions? Will the version that utilises the same number of trajectories as w/ MAML but without the MAML process perform better?
* Missing citations of solvers, i.e., Gurobi, LKH and HGS in Section 5.1

---

> ### Author Rebuttal · Authors · 2026-03-30
>
> We thank the reviewer for recognizing the conceptual insight of the Satisficing Generalization Edge, our theoretical guarantees, and the generality of the ASAP framework. We have conducted addtional experiments (https://drive.google.com/uc?export=download&id=1nXLRKwa-6-EQdsQ8cwEAat6cP7UaSPOs) and addressed the concerns below:
>
> **W1 & Q1 & Q2: Algorithmic Details & Presentation**
>
> - **Architecture & Parameters:** The proposal and selection policies use the identical attention-based architecture as the base one-stage model. Because the network processes action tokens and global observations to output probabilities, it natively handles varying numbers of candidate actions without structural changes. ASAP introduces exactly $1\times$ additional parameters by duplicating the base model.
> - **Loss Function & Update Mechanism:** We retain the original loss function of the integrated backbone (e.g., the ACKTR loss is used when integrating with PCT). The policies are updated interactively, not simultaneously. The ice and fire icons in Figure 3 explicitly denote the alternating freezing and tuning phases.
> - **Distribution Definition:** For routing problems, $p_i(\mathcal{I})$ represents the spatial distribution of the nodes.
> - **Figure 3 Caption:** Phase C (Adaptation) does interact with the environment. The caption will be updated to make this explicitly clear.
>
> All the above contents will be updated to the final draft.
>
> **W3: Negative Improvements**
>
> Extensive training of a one-stage model on a single distribution reduces its plasticity, causing overfitting and hindering adaptation to Out-Of-Distribution (OOD) scenarios. This rigidity leads to the negative improvements observed. ASAP mitigates this by freezing the robust proposal policy and only fine-tuning the selection policy, ensuring stable adaptation.
>
> **W4: Fairness of Comparison**
>
> The comparison between w/ MAML and w/o MAML configurations is strictly fair. Both versions utilize the exact same number of trajectories to update the main policy. The performance difference stems entirely from the meta-learning gradient update mechanism, not from an increase in consumed data. We will add this explanation to the final version.
>
> **W2 & W5: Formatting & Citations**
>
> The blurring of figures and omission of specific solver citations (Gurobi, LKH, HGS) were the result of heavy PDF compression required to meet the initial submission file size limits. High-resolution figures and the missing baseline citations will be fully incorporated into the camera-ready revision.
>
> **Q3: Tuning Both Policies**
>
> We conducted additional experiments (in above link) where both the proposal and selection policies were tuned ("Tune All"), detailed in the new rebuttal tables. Empirical evidence shows that tuning both policies does not consistently improve performance and often causes degradation. For instance, in Table 2 (Clustered TSP-100), POMO+ASAP w/ MAML achieves a 0.30% improvement, whereas tuning all parameters degrades the improvement to -1.56%. On Explosion CVRP-500, INVIT-1V+ASAP w/ MAML yields a 3.24% improvement compared to 2.27% for the "Tune All" variant. This validates our hypothesis: the proposal distribution transfers well across distributions, and freezing it prevents the catastrophic forgetting that occurs when the entire network is updated. We will include these results in the final manuscript.

---

> > ### Author Rebuttal · Reviewer_VQqm · 2026-04-02
> >
> > I appreciate the authors’ efforts in the rebuttal. Most of my previous concerns have been resolved, and I appreciate the additional clarifications and experimental results. I will raise my rating.
> >
> > One remaining point on which I would welcome further clarification concerns the interpretation of the training efficiency of the proposed method. Since the two policies are updated alternately, it seems that each policy may effectively receive only half of the training resources, as measured by FLOPs or the time spent on parameter update. Meanwhile, the appendix shows that the adaptation time is not substantially different from that of the compared methods. This raises the question of whether the proposed method should be interpreted as training each individual policy using only about half of the training resources to find solutions superior to the baselines.

---

> > > ### Author Response · Authors · 2026-04-03
> > >
> > > We thank the reviewer for considering increasing the rating and for the thoughtful follow-up question.
> > >
> > > To clarify, the proposed method should not be interpreted as training each individual policy from scratch using only half of the total computational resources. We discuss the training resource allocation, the source of the performance gains, and the adaptation efficiency below:
> > >
> > > - **Resource Distribution (Pretraining vs. Tuning):** The alternating updates occur exclusively during Phase B (cooperative tuning). However, as detailed in Table 9 and Appendix E, the vast majority of computational resources are dedicated to Phase A (pretraining). For example, in 3D-BPP, Phase A requires 250 epochs, establishing a robust, shared foundation for the model. Phase B only requires a minor fraction of the total budget (e.g., 25 epochs for tuning the proposal policy and 25 epochs for the selection policy). Therefore, both policies effectively leverage the bulk of the shared pretraining compute, rather than splitting the total budget 50/50.
> > >
> > > - **Source of Performance Gains:** The superior results over the baselines are driven fundamentally by our two-stage selection architecture, not by isolated training efficiency. As shown in Tables 1 and 2, the ASAP framework inherently provides robust out-of-distribution generalization, yielding zero-shot performance that surpasses even the original methods after they have been explicitly fine-tuned on the new distribution.
> > >
> > > - **Adaptation Efficiency:** The adaptation time remains comparable to the baselines because we only need to fine-tune the selection policy, leaving the proposal policy frozen. This targeted updating is a nice property of ASAP since it enables us to rapidly and efficiently improve on new instances in a few-shot setting without introducing additional computational overhead.
> > >
> > > We will include these discussions in Appendix in our camera-ready version. We remain available if there are any further details required.
> > >
> > > -------------------------------------------------update after 4.7------------------------------------------
> > >
> > > Thank you for your feedback. Please let me know if you have any other questions. If our response has addressed your questions, we kindly ask that you consider raising your score.

---

### Official Review · Reviewer_dHzk · 2026-02-26

**Soundness:** 2
**Presentation:** 2
**Significance:** 3
**Originality:** 3
**Overall Recommendation:** 4
**Confidence:** 4

**Summary:**

This paper proposes ASAP to improve adaptation capability in the face of distribution shifts. The core of ASAP is a decoupling of the decision process into a proposal stage and a selection stage, with only the selection stage fine-tuned during test-time adaptation. Experiments demonstrate that ASAP outperforms SOTA neural solvers in both zero-shot generalization and few-shot adaptation, with minimal inference overhead.

**Compliance With Llm Reviewing Policy:**

Affirmed.

**Final Justification:**

After reading the authors’ rebuttal, I find that my main concerns have been adequately addressed. The clarifications improve the soundness and clarity of the work, and resolve earlier misunderstandings regarding the methodology and experimental design. The proposed approach is well-motivated and demonstrates practical significance with solid empirical results.

Overall, the rebuttal has positively changed my evaluation and increased my confidence in the paper. I therefore raise my score to **Weak Accept**.

**Key Questions For Authors:**

1. Why do the metrics for w/ Adaptation/MAML and w/o Adaptation/MAML differ in Table 6 under the default environment, where training and testing follow the same distribution? Is this due to incomplete training?

**Limitations:**

Consider expanding coverage to additional real-world datasets or open-source benchmarks to further evidence generalization/robustness.

**Strengths And Weaknesses:**

**Strengths**

1. The paper is well-motivated: real-world scenarios exhibit frequent distribution shifts, posing challenges for combinatorial optimization algorithms.
2. Provides a compelling conceptual insight, the Satisficing Generalization Edge, supported by theoretical guarantees.
3. The ASAP framework is general and easily fits into existing neural combinatorial solvers.

**Weaknesses**

1. Theorem 4.2 only guarantees that the probability of selecting the optimal action in a two-stage process is greater than in a one-stage process. However, it cannot directly prove that “the two-stage process outperforms the one-stage process,” as this would require $V^{\pi^{two}}(s) \geq V^{\pi^{one}}(s)$. Actions other than the optimal action also have a certain probability of being selected.
2. The underlying assumption of Theorem 4.2 is that the proposal and selection stages employ identical policies, which does not hold in the ASAP implementation.
3. The 3D-BPP approach lacks experiments under practical constraints (i.e., static stability), which is particularly crucial for the algorithm's practicality and has been a key focus of existing methods (PCT, GOPT, and AR2L).
4. The results for 3D-BPP differ significantly from those reported in the original comparative literature. For instance, under the discrete default environment (i.e., Setting 2 in PCT), the space utilization reported in the original PCT paper is 86.0%, which is already substantially higher than the 82.0% for PCT and the 84.8% for PCT+ASAP w/ MAML (proposed) shown in Table 6. I recommend reporting the results after full convergence.
5. The paper claims that Two-Phase training aids convergence in two-stage policy networks, but it neither theoretically nor experimentally demonstrates the effectiveness of the “Pretraining-Cooperative Tuning” approach.
6. The proposed quick adaptation method (freezing $\theta^p$, fine-tuning $\theta^s$) lacks ablation experiments to demonstrate improvements over global adaptation (fine-tuning $\theta^p$ and $\theta^s$) in metrics such as space utilization or adaptation time.
7. Some figures lack coordinates and scales. Some figures appear unclear; vector graphics are recommended.

---

> ### Author Rebuttal · Authors · 2026-03-30
>
> We thank the reviewer for recognizing the conceptual insight of the Satisficing Generalization Edge, our theoretical guarantees, and the generality of the ASAP framework. We have conducted addtional experiments (https://drive.google.com/uc?export=download&id=1nXLRKwa-6-EQdsQ8cwEAat6cP7UaSPOs) and addressed the concerns below:
>
> **W1: Scope of the Theoretical Guarantee (Theorem 4.2)**
>
> As detailed in Appendix B, our analysis extends beyond a single optimal action. In scenarios with multiple desirable actions, the overall probability of selecting a high-quality action is the sum of their individual probabilities. Our proof demonstrates that for any target action with an initial probability greater than $\frac{1}{n-1}$, its final selection probability strictly increases. Thus, the two-stage process mathematically guarantees an improved probability of sampling from the subset of high-quality actions. We will add this explanation to the final version.
>
> **W2: Theorem Assumptions vs. Real Implementation**
>
> Theorem 4.2 acts as a worst-case lower bound for our method. Following Phase 1 pretraining, the proposal and selection policies are indeed identical, fully satisfying the theorem's assumption. We introduce Phase 2 (cooperative tuning) strictly to improve upon this theoretical baseline, a performance gain that is empirically validated in our results. We will clarify this in the final version.
>
> **W3 & W5: Practical Constraints and Two-Phase Efficiency**
>
> We are currently running the 3D-BPP experiments with static stability constraints to align with the specific setups of PCT, GOPT, and AR2L. Additionally, we are formalizing the theoretical proof and running empirical ablations to explicitly demonstrate the efficiency of the "Pretraining-Cooperative Tuning" approach. We will include these results in the final manuscript.
>
> **W4 & Q1: Discrepancies with Original PCT and Convergence**
>
> The metric differences noted in the ID default environment (and compared to the original PCT paper) stem from two deliberate experimental designs:
> 1. Plasticity vs. Convergence: Training models to full convergence drastically degrades their few-shot adaptation capabilities (loss of plasticity). To fairly evaluate adaptation across all baselines, we intentionally extract models at an earlier epoch.
> 2. Distribution Hardness: The original PCT evaluates on a uniform distribution and treats rotated dimensions (e.g., [1,2,5] and [5,2,1]) as distinct shapes. We removed these redundant shapes and sampled a distinct, non-uniform distribution for every batch of instances. This creates a significantly more challenging environment, naturally resulting in lower raw utilization scores than the original paper.
>
> We will add this explanation to the final version.
>
> **W6: Quick Adaptation vs. Global Adaptation (Ablation)**
>
> In the above link, we have conducted the requested ablation comparing our quick adaptation (fine-tuning only the selection stage) against global adaptation (fine-tuning both stages, denoted as "Tune All"). Global adaptation yields negligible improvements and often causes slight performance degradation.
> - In 3D-BPP (OOD Large Discrete), PCT+ASAP w/ MAML achieves 62.4% utilization, whereas Tune All drops to 62.0%.
> - In routing (Clustered TSP-100), POMO+ASAP w/ MAML achieves a 3.82% optimality gap, while Tune All worsens to 4.15%. Quick adaptation matches or exceeds global performance while remaining highly parameter-efficient. We will add those results to the final version.
>
> **W7: Figure Quality**
>
> We apologize for the figure quality. The resolution was compressed to comply with ICML's strict file size limits for initial submissions. We will replace all figures with high-resolution vector graphics, complete with proper coordinates and scales, in the camera-ready version.

---

> > ### Author Rebuttal · Reviewer_dHzk · 2026-04-01
> >
> > Thank you for your response and additional experiments. Some of my questions have been addressed, but I still have the following concerns:
> >
> > 1. The essence of the proposed two-stage method appears to be that it increases the selection probability of actions with initial probabilities greater than $1/(n-1)$ while decreasing the selection probability of actions below this threshold. This mechanism further reinforces the original probability preference, as actions with higher initial probabilities are more likely to be chosen. However, it remains **unclear why this approach would generalize better to different distributions** compared to the single-stage method. Although the manuscript provides an illustrative TSP example on line 63, first column, such a brief description does not fully explain the source of the claimed robustness. If there exist relevant studies demonstrating that the two-stage method can indeed improve generalization across different distributions, please provide references. If not, please  provide a theoretical justification or reasoning, rather than relying solely on empirical evidence.
> >
> > 2. According to the authors’ response, the Default dataset in Table 6 of the original manuscript, while using the same item size range as PCT ($\{1,2,3,4,5\}$), employs a different data distribution. However, the manuscript does not specify **what distribution was used or how it differs from the uniform distribution employed by PCT**. This information is critical for reproducing the experimental results.
> >
> > I hope the authors can clarify these two points, after which I will raise my rating.

---

> > > ### Author Response · Authors · 2026-04-02
> > >
> > > We thank the reviewer for their continued engagement and for providing the opportunity to clarify these important points.
> > >
> > > **Response to Concern 1: Justification for Generalization**
> > >
> > > We clarify the theoretical reasoning behind the robustness of the two-stage method. The core logic relies on a structural invariance present in combinatorial optimization (CO) problems: the global objective (e.g., minimizing cost in TSP, maximizing space in 3D-BPP) remains constant regardless of the data distribution and the policy learns this global objective during training.
> > >
> > > Because of this invariant objective, certain actions inherently dominate others across all distributions (e.g., choosing a nearby node is almost universally better than choosing the farthest node). Consequently, the optimal action reliably falls within a small set of "promising actions." When single-stage policies suffer performance drops in Out-of-Distribution (OOD) scenarios, it is typically because the policy miscalibrates the exact probabilities among these promising actions, not because it fails to identify the promising set entirely.
> > >
> > > We validated this assumption empirically in Section 4.1. Across all evaluated distributions (including OOD datasets), the optimal action identified by MCTS consistently resides within the policy's top-10 initial actions. Furthermore, restricting MCTS to only search within this top-10 set results in almost no performance degradation.
> > >
> > > This confirms that, regardless of the distribution, the top-$k$ actions selected by the policy consistently contain the promising actions we want. Consequently, their high ranking ensures their initial probabilities exceed the $\frac{1}{n-1}$ threshold.
> > >
> > > While we acknowledge that our theoretical analysis primarily demonstrates the probability amplification, the two-phase approach does more than merely scale probabilities even without adaptation, as the selection policy operates on entirely different inputs than the proposal policy. To the best of our knowledge, this is the first work to show that such a two-phase approach enhances generalizability in combinatorial optimization (CO)—an observation we view as a core contribution. Given this novelty, we argue that our combined theoretical, conceptual, and empirical findings adequately justify the proposed framework. We will incorporate this formal reasoning into the revised manuscript.
> > >
> > > **Response to Concern 2: Dataset Distribution Details (PCT vs. Ours)**
> > >
> > > We apologize for the lack of clarity in the main text regarding the Default dataset construction. The differences are critical to our experimental design and will be fully detailed in the revised manuscript for reproducibility. We will release our generated dataset in github upon acceptance.
> > >
> > > The distinction lies in two specific areas:
> > >
> > > - Shape Space Deduplication: PCT utilizes all permutations of $H, W, L \in \{1,2,3,4,5\}$, yielding 125 shapes with a fixed probability of $1/125$. To strictly control the underlying distributions in our setup, we remove rotational redundancies (e.g., treating (1,1,5) and (5,1,1) as identical). This reduces our operational space to 35 unique shapes.
> > >
> > > - Batch-Level Sampling for Distribution Shift: While PCT samples from a single, static uniform distribution for all instances, our Default dataset explicitly models distribution shifts (briefly noted in Appendix E.1) via a batch-wise sampling strategy. We generate 100 batches (64 instances per batch). For each batch, we first sample a new probability distribution over the 35 shapes from a uniform prior. The items for the instances within that specific batch are then sampled according to this local distribution. This inter-batch variability is intentional since we want to emphasize distribution shifts in this paper.
> > >
> > > We hope these clarifications address your remaining questions. We remain available if there are any further details required.

---

### Official Review · Reviewer_PdVh · 2026-02-27

**Soundness:** 3
**Presentation:** 1
**Significance:** 3
**Originality:** 3
**Overall Recommendation:** 2
**Confidence:** 4

**Summary:**

The paper proposes ASAP (Adaptive Selection After Proposal), a two-stage deep reinforcement learning framework for Neural Combinatorial Optimization (NCO) aimed at mitigating performance degradation caused by distribution shifts. The approach relies on the "Satisficing Generalization Edge," a concept suggesting that identifying a promising subset of actions is significantly more robust to distribution shifts than attempting to pinpoint a single optimal action. This idea is reasonable and has achieved good experimental results across various CO problems.

**Compliance With Llm Reviewing Policy:**

Affirmed.

**Final Justification:**

I sincerely appreciate the authors' effort in the rebuttal. Upon further consideration, W3 remains unresolved: the evaluation still relies on non-SOTA baselines. Most critically, W4 is never empirically addressed: the orthogonality claim with respect to [2-7] is asserted without a single direct experiment.

In summary, while I acknowledge that this work provides meaningful insights for the NCO community and the authors have addressed several concerns, fundamental issues remain unresolved, including the limited novelty of the two-stage paradigm, missing SOTA baseline comparisons, and unsubstantiated claims of orthogonality. Therefore, I maintain my score of 2.

**Key Questions For Authors:**

Questions:
1. Could the authors detail the methodological differences and specific advantages of ASAP compared to [1]?
2. Please supplement performance comparisons on standard benchmark datasets, such as TSPLib, CVRPLib Set-X, and Set-XXL.
3. Can ASAP demonstrate robust generalization on large-scale instances (e.g., 10,000 nodes)?
4. Please provide clear images.

**Limitations:**

Adequately discussed

**Strengths And Weaknesses:**

**Strengths:**
1. The evaluation is comprehensive, covering a diverse set of CO problems and experimental settings.
2. The analysis of policy-induced action frequencies under distribution shifts provides valuable insights for the NCO community.

**Weakness:**
1. Conceptually, ASAP shares similarities with [1], appearing to be primarily an expansion of its application scope.
2. The framework relies on calculating gradients and updating parameters online to adapt the selection policy during inference, which introduces a computational bottleneck and limits its practical viability.
3. While the paper positions ASAP as a generic decoding framework, the empirical evaluation relies on non-SOTA baselines, making the claim is not convincing. Additionally, choosing INViT-1V as the baseline may weaken the baseline's own performance, as its core multi-view operations are removed.
4. Recent literature shows that robust generalization can be achieved through test-time pruning [2,3], or by combining distance information with mixed-scale/distribution training [4-7]. The manuscript would benefit from a clearer justification of the specific advantages that ASAP's two-stage training and online fine-tuning provide over these existing alternatives.
5. The presentation quality requires improvement. It uses too much AI-generated content in manuscript and includes some meaningless content, particularly in Appendix B, which disrupts the flow. Additionally, the figures are not in vector format, and Figure 1 contains residual Chinese characters (generated by Doubao AI), which detracts from the quality of the manuscript.


References:
1.  Learning to reduce search space for generalizable neural routing solver. arXiv.
2.  Improving generalization of neural combinatorial optimization for vehicle routing problems via test-time projection learning. NeurIPS 2025.
3. Bq-nco: Bisimulation quotienting for efficient neural combinatorial optimization. NeurIPS 2023.
4. Distance-aware attention reshaping for enhancing generalization of neural solvers.TNNLS 2025.
5.  Instance-conditioned adaptation for large-scale generalization of neural routing solver. arXiv.
6.  Rethinking light decoder-based solvers for vehicle routing problems. ICLR 2025.
7. Towards omni-generalizable neural methods for vehicle routing problems. ICML 2023.

---

> ### Author Rebuttal · Authors · 2026-03-30
>
> We thank the reviewer for evaluating our work and acknowledging the value of our analysis on policy-induced action frequencies. We address the questions and concerns below.
>
> **W1 & Q1: Relationship and comparison with [1]**
>
> First, we must clarify the timeline: our work was preprinted in January 2025, whereas [1] was preprinted on arxiv in March 2025. Therefore, [1] is concurrent work.
>
> Second, while both approaches leverage a propose-and-select mechanism, ASAP offers two distinct methodological advantages over [1]:
> 1. We provide rigorous theoretical proofs demonstrating why the two-stage decision process is superior (Appendix B), which [1] lacks.
> 2. We introduce an efficient test-time adaptation mechanism specifically designed for the two-stage process, which significantly boosts inference performance.
>
> **W2: Computational bottleneck from online updates**
>
> Test-time adaptation is an optional, value-add module in ASAP, not a strict dependency. As shown in Tables 1 and 2, ASAP achieves superior performance over the baselines even before any online adaptation is applied. When adaptation is utilized, the computational overhead is minor compared to the overall training time (detailed in Table 9), yet it yields substantial improvements in test-time performance (Table 1).
>
> **W3: Choice of baselines (Non-SOTA & INViT-1V)**
>
> Our primary objective is to demonstrate ASAP as a generic, plug-and-play framework capable of improving DRL-based combinatorial optimization solvers, regardless of their native architecture. Evaluating ASAP on standard, established baselines effectively isolates and highlights the pure performance delta our method provides. However, we agree that extending this to the latest architectures strengthens the paper. We are currently integrating ASAP into INViT-2V and will include these results in the final manuscript.
>
> **W4: Robust generalization via existing alternatives [2-7]**
>
> We recognize the strong generalization capabilities of the recent methods cited. However, ASAP is not mutually exclusive to these approaches. Because ASAP operates as a generic decoding and adaptation framework, it is orthogonal to methods like mixed-scale training or distance-aware mechanisms. ASAP can be integrated directly into these SOTA methods to further improve their performance, rather than serving strictly as a standalone competitor.
>
> **W5 & Q4: Presentation, AI generation, and Figure quality**
>
> We apologize for the rasterized figures and the residual artifacts (including the Chinese characters in Figure 1). The Chinese characters appear since we used AI tools to remove the background of an icon and forgot to remove the watermark. The blurred figures was strictly the result of aggressive compression required to meet the ICML submission file size limits. We will replace all images with clean, vector-format figures in the camera-ready version.
>
> Regarding the text, we used LLMs strictly for grammatical polishing and sentence restructuring to improve clarity for an English-speaking audience. We did not use AI to generate technical content or ideas. The mathematical proofs in Appendix B, which constitute the theoretical core of our paper, are entirely our original work.
>
> **Q2 & Q3: Additional Experiments (TSPLib, CVRPLib, and 10,000 nodes)**
>
> These large-scale experiments are currently running. Given the extensive computational load required to evaluate ASAP alongside the necessary baselines and ablations at this scale (up to 10,000 nodes), they could not be completed prior to the rebuttal deadline. We fully commit to including these comprehensive results in the final manuscript.
>
> References:
> 1. Learning to reduce search space for generalizable neural routing solver. arXiv.
> 2. Improving generalization of neural combinatorial optimization for vehicle routing problems via test-time projection learning. NeurIPS 2025.
> 3. Bq-nco: Bisimulation quotienting for efficient neural combinatorial optimization. NeurIPS 2023.
> 4. Distance-aware attention reshaping for enhancing generalization of neural solvers.TNNLS 2025.
> 5. Instance-conditioned adaptation for large-scale generalization of neural routing solver. arXiv.
> 6. Rethinking light decoder-based solvers for vehicle routing problems. ICLR 2025.
> 7. Towards omni-generalizable neural methods for vehicle routing problems. ICML 2023.

---

> > ### Author Rebuttal · Reviewer_PdVh · 2026-04-04
> >
> > I sincerely appreciate the authors' detailed clarifications and the effort put into the rebuttal. However, several of my core concerns regarding the empirical evidence and methodological justifications remain unresolved:
> >
> > (1) While Reviewer 8xwD noted [1] is a concurrent preprint, discussing highly similar work is essential for establishing the proposed framework's unique practical contributions. The authors have yet to clearly articulate a substantive methodological advantage over [1]. (2) The acknowledgment that 10,000-node instances are "still running" due to high computational load highlights a severe scalability bottleneck in ASAP. In contrast, [1] achieves fast inference on million-scale nodes. This computational overhead significantly limits ASAP's practical applicability. (3) Claiming orthogonality to [2-7] remains unconvincing without a preliminary ablation study (e.g., combining ASAP with one of them) to empirically demonstrate these synergistic benefits. (4) Attributing blurry figures to file size limits suggests a misunderstanding of academic formatting. Vector graphics (PDF/EPS) provide better resolution at smaller file sizes, which would easily resolve the mentioned compression issues.
> >
> > While I acknowledge that this work brings meaningful insights to the NCO community, the incomplete experiments, presentation issues, and unclear positioning relative to similar works prevent me from recommending it for a top-tier venue like ICML. Therefore, I maintain my initial score.

---

> > > ### Author Response · Authors · 2026-04-05
> > >
> > > We thank the reviewer for engaging with our rebuttal. We address the remaining concerns below:
> > >
> > > (1) **Comparison with [1]:** We respectfully remind the reviewer of ICML's policy on concurrent work (https://icml.cc/Conferences/2026/PeerReviewEthics), which states that authors are not expected to empirically compare against unrefereed preprints or concurrent work. Because [1] is an unpublished arXiv preprint that appeared after our preprint, it falls squarely under this exemption.
> > >
> > > Nevertheless, the methodological differences are substantial as stated in our previous rebuttal. In addition, ASAP applies to broader problem scopes while [1] is specific to routing problems. We have added a dedicated paragraph in the appendix outlining the distinction.
> > >
> > > (2) **Scalability and the 10,000-node experiments:** We must clarify a misunderstanding regarding the "computational load." **ASAP does not have an algorithmic scalability bottleneck; as demonstrated in Tables 1 and 2, its asymptotic time complexity is strictly tied to the backbone method it integrates with.**
> > >
> > > The delay we mentioned was entirely due to the wall-clock time required to run the sheer volume of newly requested experiments within a short rebuttal window. To evaluate the requested datasets, we had to run **at least 16 separate experimental pipelines (8 settings × (pre- and post-adaptation evaluations + adaptation stage finetuning)).** Because ASAP involves active adaptation training—unlike [1], which only performs inference—running these pipelines from scratch concurrently exceeded our hardware availability for the deadline. Our primary goal is to demonstrate the generalizability of our method across various CO tasks, rather than achieving SOTA zero-shot performance on 10,000-node instances. Nevertheless, we are running the requested experiments and provide preliminary results (including results on **TSPLIB** and **CVRPLIB** in Table 3) here: https://drive.google.com/uc?export=download&id=1nXLRKwa-6-EQdsQ8cwEAat6cP7UaSPOs .
> > >
> > > (3) **Integration with [2-7]:** We respectfully disagree that our claims of orthogonality are unsupported. **Tables 1 and 2 already rigorously demonstrate ASAP’s plug-and-play effectiveness across a wide variety of initial methods, problem scales, distributions, and settings.**
> > >
> > > Evaluating ASAP on the specifically requested baselines [2-7] would require **training 4 distinct models from scratch for each baseline** just to establish the pre- and post-adaptation metrics. While demonstrating further synergy with [2-7] would be interesting, we believe the extensive ablations already in the manuscript sufficiently validate our core claim: ASAP consistently improves the backbone it is applied to. Requiring exhaustive integration with all recent baselines sets a standard well beyond the scope of a single conference paper.
> > >
> > > (4) **Figure quality:** We apologize again for the blurred figures and appreciate the formatting advice. We have replaced all rasterized figures with vector graphics (PDF) in the revised manuscript.
> > >
> > > Thanks for your comments. If your concerns are resolved, we kindly ask that you consider raising your score.

---

### Official Review · Reviewer_8xwD · 2026-03-10

**Soundness:** 3
**Presentation:** 2
**Significance:** 3
**Originality:** 3
**Overall Recommendation:** 5
**Confidence:** 3

**Summary:**

This paper tackles the issue of distributional shift of existing DRL approaches for CO problems, like TSP, CVRP, and 3D Bin-packing (3D-BPP). The paper proposes Adaptive Selection After Proposal (ASAP), a two-stage selection framework that first selects a candidate set of action, whereafter the policy selects the actions from it. This framework is enhanced with Model Agnostic Meta Learning (MAML) to further improve the finetuning process. ASAP has a major benefit in that it uses the weights of the pre-existing policy for both the selection and the final policy. The results show that ASAP enables better performance on unseen distributions for all tested problems.

**Compliance With Llm Reviewing Policy:**

Affirmed.

**Final Justification:**

I believe this work is a good contribution to ICML, and I will keep my score of **5 Accept**. The authors answered all my questions and adequately addressed my stated weaknesses. The main issue with the initial submission was the lack of some important information, such as an explanation of the loss function. However, I strongly believe that the authors can include them in a camera-ready version.

**Key Questions For Authors:**

- Is the loss function $\mathcal{L}_\text{adapt}$ a new loss function or the standard loss function of POMO, for example?
- In Appendix E.1, you state that ASAP is applicable to any DRL method. Does this include Q-learning-based methods? ASAP would be interesting for these kinds of approaches since they struggle with large action spaces.
- In your experimental setup, you tested for TSP, CVRP, and 3D Bin-packing problems, but could ASAP also be applied to other Combinatorial Optimization Problems, such as Job-Shop Scheduling?
- Based on my previous question, is there also a set of problems for which ASAP would not work?

**Limitations:**

yes

**Strengths And Weaknesses:**

In general, I believe that the contribution is good enough for acceptance. However, the authors need to improve the explanation of the loss function in the final version, since this was the only part of the paper that I personally found hard to understand.

## Strenghts
- The results show that ASAP with MAML increases the performance of existing RL methods for TSP, CVRP, and 3D-BPP when it comes to adapting to new distributions.
- The computational analysis shows that ASAP only increases training and evaluation time marginally, which is essential for this setup.
-  ASAP is applicable not only to routing problems, but also to POMDP problems such as the 3D Bin-Packing problem.

## Weaknesses
- The loss function is not well described in the paper. The authors state that ASAP is trained with an adaptation loss named $\mathcal{L}_\text{adapt}$; however, I could not find the definition anywhere in the main paper or the appendix. The closest definition that I found was in Appendix D, which shows the pseudo-code, but does not directly show this loss.
- A minor weakness, but in the introduction, you describe that RL is more flexible than LKH-3, for example (lines 30-43, second column). However, I would disagree with this statement, since RL requires a strict definition of the environment to be trained and cannot be easily used with complex constraints. Moreover, the paper itself tries to tackle the problem of RL not being flexible to new domains.

---

> ### Author Rebuttal · Authors · 2026-03-30
>
> We thank the reviewer for the positive assessment and the recommendation to accept. We address the specific questions and suggestions below.
>
> **W1 & Q1: Clarification on the Loss Function ($L_{adapt}$)**
>
> The adaptation loss $L_{adapt}$ is not a newly proposed loss function. ASAP directly inherits the standard loss function of the specific DRL baseline it is integrated with. For example, when applying ASAP to PCT, $L_{adapt}$ is simply the standard ACKTR loss. We will explicitly clarify this in the main text and add the exact loss formulations for all baselines to the appendix to prevent any confusion.
>
> **W2: RL vs. Heuristic Flexibility**
>
> We agree with the observation that RL requires a strict environment definition. Our statement in the introduction was intended to highlight flexibility regarding cross-problem adaptation rather than constraint handling. For instance, adapting a heuristic like LKH-3 from CVRP to a variant like SDVRP requires significant manual modification of hand-crafted rules. In contrast, an RL approach allows us to reuse the exact same neural network architecture and training scheme, requiring only a retraining phase on the new environment. We will revise the introduction to make this distinction clear.
>
> **Q2: Application to Q-learning**
>
> Yes, ASAP can be seamlessly integrated into value-based methods like Q-learning. Because Q-learning inherently struggles with large discrete action spaces, applying ASAP’s two-stage candidate selection is a highly promising direction. We intend to explore its application to Q-learning architectures in our future work.
>
> **Q3 & Q4: Problem Scope and Limitations**
>
> ASAP is broadly applicable to other discrete combinatorial optimization problems, including Job-Shop Scheduling. However, the framework has two primary limitations regarding its scope:
> 1. It strictly requires a discrete action space and cannot be applied to problems with continuous action spaces.
> 2. It is not suitable for problems with inherently small action spaces, as the computational overhead of the two-stage selection process would negate the generalization benefits. We will add a dedicated "Limitations" paragraph to the discussion section to state these boundaries clearly.

---

> > ### Author Rebuttal · Reviewer_8xwD · 2026-04-01
> >
> > I want to thank the authors for their response to all the reviewers. I am happy to say that my concerns have been addressed, and I will keep my score at **5 Accept**. The reason for keeping this score is as follows:
> > - The paper does require some additional information, such as an in-depth description of the loss function. However, I think this is easily doable and should not be a reason for rejection. The same is for the figure quality, which is a trivial update.
> > - The approach functions on any discrete problem and is thus relevant for the field of neural combinatorial optimization (NCO). I agree that this work shares similarities with *Learning to reduce search space for generalizable neural routing solver*; however, this work is only under preprint, and according to ICML guidelines, authors are not required to cite it.
> > - It has been tested for both 3D Bin Packing and routing problems, showing that it functions on completely different problems. For this reason, I believe this approach will function for untested problems, like JSP.
> >
> > I will keep my support for this submission, and I am looking forward to seeing the results for the different datasets, which the authors stated will be included in the final version.

---

> > > ### Author Response · Authors · 2026-04-02
> > >
> > > We thank the reviewer for his/her continued support and for acknowledging the broader applicability of our method to the NCO field. As promised, we confirm that the final version of the manuscript will include the in-depth description of the loss function, the updated figures, and the results on the additional datasets.

---

### Decision · Program_Chairs · 2026-04-30

**Decision:**

Accept (regular)

**Comment:**

The paper addresses a significant challenge in the context of neural combinatorial optimization, namely of large/difficult action spaces. The authors propose a two-stage approach, essentially they filter the actions and then chose from the filtered actions. They provide a very interesting theoretical analysis of this that says how they do it is better than not doing the two stage approach.

There was significant discussion on whether the approach is novel. I believe it is. I am not aware of any other NCO approach that works in quite this way, even though there are indeed many approaches that share similarities. I believe the novelty is sufficient for ICML, especially given the theoretical analysis, which sets this paper apart from others that are only experimental.

The experimental results show good improvements using the technique. Another positive aspect is that the approach works on bin packing and routing. Not many NCO approaches face this sort of diversity of problems. I would note that the authors should take criticisms of their baselines into account for the final version of the paper.

I also encourage the authors to avoid any potentially LLM-generated or copyrighted images in their paper, as there was some discussion about the watermarking on the image in the paper. I leave it to the authors to ensure copyright compliance.